# DisC-GS: Discontinuity-aware Gaussian Splatting

**Haoxuan Qu** [*]
Lancaster University
U.K.
h.qu5@lancaster.ac.uk

**Zhuoling Li** [*]
Lancaster University
U.K.
z.li81@lancaster.ac.uk

**Hossein Rahmani**
Lancaster University
U.K.
h.rahmani@lancaster.ac.uk

**Yujun Cai**
University of Queensland
Australia
vanora.caiyj@gmail.com

**Jun Liu** [†]
Lancaster University
U.K.
j.liu81@lancaster.ac.uk

## Abstract

Recently, Gaussian Splatting, a method that represents a 3D scene as a collection of Gaussian distributions, has gained significant attention in addressing the task of novel view synthesis. In this paper, we highlight a fundamental limitation of Gaussian Splatting: its inability to accurately render discontinuities and boundaries in images due to the continuous nature of Gaussian distributions. To address this issue, we propose a novel framework enabling Gaussian Splatting to perform discontinuity-aware image rendering. Additionally, we introduce a Bézier-boundary gradient approximation strategy within our framework to keep the "differentiability" of the proposed discontinuity-aware rendering process. Extensive experiments demonstrate the efficacy of our framework.

## 1 Introduction

Novel view synthesis aims to generate images accurately from novel viewpoints in a captured 3D scene. Its significance spans across diverse applications, such as autonomous driving [45], virtual reality [14], and 3D content generation [42]. Recently, for better tackling novel view synthesis, Neural Radiance Field (NeRF) [36] and a variety of NeRF-based methods [3, 4] have been proposed, which represent 3D scenes in an implicit manner as neural radiance fields. However, their general reliance on a heavy volume rendering mechanism often results in slow rendering speeds [30, 13], limiting their practicality across real-world applications. While some methods [18, 19] have proposed to accelerate the rendering process of NeRF from different perspectives, they often achieve this at the expense of noticeably compromising the quality of the generated images [49], which is evidently undesirable.

More recently, Gaussian Splatting [30], which explicitly represents the 3D scene as a collection of Gaussian distributions, has been proposed as an appealing alternative to NeRF. Specifically, rather than generating novel-view images through the time-consuming process of volume rendering, Gaussian Splatting enables images from novel viewpoints to be generated by simply splatting (projecting) [53, 54] these Gaussian distributions onto the image plane. By doing so, Gaussian Splatting achieves real-time rendering of novel-view images, while maintaining its rendered images to be of competitively high visual quality compared to NeRF-rendered ones. Due to its compelling capability, Gaussian Splatting has received lots of research attention [26, 13, 42, 49, 12, 21, 25].

---

[*]Both authors contributed equally to the work.
[†]Corresponding Author

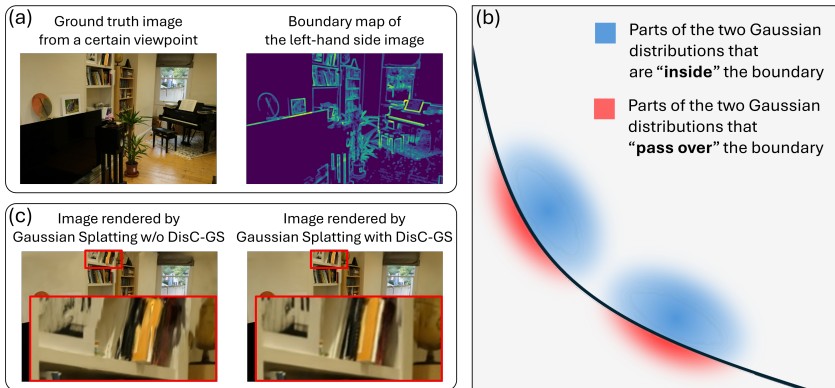

Figure 1: (a) Illustration of a ground truth image, containing numerous discontinuities and boundaries, that is expected to be rendered from a certain viewpoint of a 3D scene. We generate the boundary map in (a) utilizing the Canny algorithm [9]. (b) Illustration of Gaussian distributions projected onto the image plane. As shown, since Gaussian distributions are continuous, they can inevitably "pass over" the (hard) boundary represented by the curve. (c) Illustration of images rendered with and without applying DisC-GS. As shown, without DisC-GS, Gaussian Splatting can fail to accurately render boundaries. In contrast, applying DisC-GS ensures that boundaries and discontinuities in the image are properly rendered. **More qualitative results are in Appendix B.** (Best viewed in color.)

However, in this paper, we argue that Gaussian Splatting may still be sub-optimal in accurately synthesizing novel views, due to its inherent weakness in representing (rendering) *discontinuities and boundaries* with its collection of *continuous* Gaussian distributions. Specifically, due to the general complexity of 3D scenes, the expected image to be rendered often contains numerous *discontinuities and boundaries* (as shown in Fig. 1(a)). However, Gaussian Splatting represents each of its generated images using only *continuous* Gaussians projected onto the image plane. Considering this, as illustrated in Fig. 1(b), the inherent continuity of Gaussian distributions can result in some parts of the distribution inevitably "passing over"("spilling over") the boundaries of sharp features in the image. This can lead to Gaussian Splatting rendering the sharp boundaries in the image with blurriness (as shown in Fig. 1(c)), which can significantly reduce the quality of the rendered image.

Based on the above argument, in this paper, we aim to enable Gaussian Splatting to bypass its original intrinsic weakness, and render discontinuities and boundaries properly. However, this can be non-trivial owing to the following challenges: (1) Since 3D scenes generally can be complex, as illustrated in Fig. 1(a), various different kinds of boundaries with diverse shapes can all exist in the image rendered from a certain 3D scene. Thus, it can be difficult to represent and render these diverse boundaries properly and seamlessly in a Gaussian-Splatting-based framework. (2) Meanwhile, recall that continuity serves as the prerequisite for a function to be differentiable. Thus, during the process of learning the 3D scene representation using Gaussian Splatting, how to maintain the "differentiability" of the process in existence of discontinuities, i.e., enabling the loss calculated over the rendered images that contain "discontinuous" (sharp) boundaries to properly guide the learning of the 3D scene representation, is also challenging. To handle the above challenges, in this work, we propose **DisC**ontinuity-aware **G**aussian **S**platting (**DisC-GS**), a novel framework that can for the first time, enable Gaussian Splatting to represent and render discontinuities properly in its image rendering process, which handles a key limitation of the original Gaussian Splatting technique. We illustrate the rendering process of our framework in Fig. 2, and outline our framework as follows.

Overall, to enable Gaussian Splatting to properly render discontinuities and boundaries, our framework introduces a "pre-scissoring" step. Specifically, for each Gaussian distribution representing the 3D scene, rather than directly rendering its entire 2D projection on the image plane, we first segment ("pre-scissor") the projected Gaussian distribution along the specified boundaries. However, achieving this requires representing boundaries with various shapes accurately. Here, we get inspiration from that, the cubic Bézier curve, conveniently represented by a group of four control points, has shown capable of efficiently parameterizing curves of various shapes with low computational complexity [16, 34]. Considering this, in our framework, we aim to use the cubic Bézier curves to represent boundaries. Specifically, we first introduce each Gaussian distribution representing the 3D scene with an additional attribute, which when projected onto the image plane, can serve as the control points of

the cubic Bézier curves. After that, given a viewpoint, based on the control points projected onto the image plane corresponding to the viewpoint, we use the cubic Bézier curves formulated from these control points to represent the desired boundaries w.r.t. each projected Gaussian distribution. Finally, leveraging the derived boundaries, we can achieve discontinuity-aware image rendering through a modified $\alpha$-blending function (as discussed in Sec. 4.1).

Through the above process, we can render discontinuities and boundaries successfully (in the forward direction). However, the above process alone cannot be seamlessly integrated into the Gaussian Splatting pipeline. This is because, owing to the incorporation of the boundary information, the modified $\alpha$-blending function now is no longer continuous everywhere. This can cause Gaussian Splatting, naively integrated with the above process, to become non-differentiable, and thus results in difficulties during the learning process of the 3D scene representation. To tackle this problem, in our framework, we further introduce a Bézier-boundary gradient approximation strategy, by which during backpropagation, we can enable gradients to properly pass through the modified $\alpha$-blending function, and thus keep our framework to be still "differentiable". With the above designs properly involved, our DisC-GS framework can finally enable Gaussian Splatting to render discontinuities and boundaries properly, seamlessly addressing its original key intrinsic limitation.

The contributions of our work are summarized as follows. 1) We proposed DisC-GS, an innovative framework for the novel view synthesis task. To the best of our knowledge, this is the first effort that enables Gaussian Splatting to represent and render boundaries and discontinuities properly in its image rendering pipeline, which tackles a key intrinsic limitation of Gaussian Splatting. 2) We introduce several designs in our framework to enable it to render images in a discontinuity-aware manner, while also to keep its "differentiability" in the presence of discontinuities. 3) DisC-GS achieves superior performance on the evaluated benchmarks.

## 2 Related Work

**Novel View Synthesis.** Owing to the wide range of applications, the task of novel view synthesis has received lots of research attention [23, 40, 39, 43, 24, 36, 3, 4, 5, 44, 50, 7, 11, 18, 19, 37, 30, 26, 13, 49, 20, 46, 22, 35, 33, 32, 48, 52, 17]. In the early days, with the emergence of CNN, different works have been proposed to leverage CNN in this task from different perspectives. Among them, Hedman et al. [23] proposed to use CNN to predict blending weights, and Sitzmann et al. [40] proposed to seek help from CNN in performing volumetric ray-marching. As time passed, NeRF tends to become a popular way in representing 3D scenes. Specifically, the original version of NeRF is first proposed by Mildenhall et al. in [36] and after it comes out, a variety of different NeRF-based methods have been further proposed, such as Mip-NeRF [3], NeRF++ [50], and Point-NeRF [44]. Despite the increased efforts, a weakness of NeRF-based methods can be that, to render novel-view images in high visual quality, they often still require a slow rendering process [30, 13]. This can negatively affect the usage of these methods in many real-world scenarios.

Considering this, more recently, the Gaussian Splatting technique, which can render novel-view images in good quality while at the same time in real-time speed, as an attractive alternative to NeRF, has gained plenty of research attention. Specifically, Kerbl et al. [30] proposed to represent a 3D scene as a collection of 3D Gaussian distributions and made the first attempt to perform novel view synthesis using the Gaussian Splatting technique. After that, Huang et al. [26] pointed out that representing the 3D scene utilizing 3D Gaussian distributions can lead to a viewpoint inconsistency problem. To tackle this problem, they proposed to represent the 3D scene with 2D Gaussian distributions instead. Moreover, Cheng et al. [13] proposed to seek help from the classical patch matching technique to better guide the densification of Gaussian distributions, and Zhang et al. [49] formulated a new loss function in the frequency space to better regularize the learning process of Gaussian Splatting.

Different from these existing Gaussian-Splatting-based methods that typically render complete Gaussian distributions during the image rendering process, we here argue that a key limitation of the original Gaussian Splatting technique lies in that, directly rendering the complete Gaussian distributions can lead boundaries and discontinuities in the image to be inaccurately rendered. Considering this, in this work, we propose to enable Gaussian distributions to be "pre-scissored" along desired boundaries before rendered. This for the first time, enables Gaussian Splatting to represent and render discontinuities and boundaries properly.

**Curve Representation.** The idea of representing a curve in a parametric way has been studied in various tasks [34, 27, 38, 16, 28, 8], such as lane detection [16], trajectory prediction [27], and text spotting [34]. Here in this work, we design a novel framework, which enables Gaussian Splatting to perform discontinuity-aware novel-view image rendering, via utilizing the cubic Bézier curves to parametrically contour the boundaries in the image plane.

# 3 Preliminary

**Gaussian Splatting.** Gaussian Splatting represents the 3D scene explicitly as a collection of anisotropic Gaussian distributions. In specific, in the collection, each Gaussian is defined with the following attributes: (1) its center $\mu \in \mathbb{R}^3$, (2) its covariance matrix $\Sigma \in \mathbb{R}^{3 \times 3}$, (3) its spherical harmonic (SH) coefficients $c_{SH} \in \mathbb{R}^{3 \times (k+1)^2}$ representing its color from different viewpoints (where $k$ denotes the order of SH), and (4) its opacity $\alpha \in \mathbb{R}^1$. Regarding the covariance matrix $\Sigma$, it is important to ensure $\Sigma$ remains positive semi-definite during the learning process of the 3D scene representation. To achieve this, $\Sigma$ is expressed as $\Sigma = RSS^T R^T$, where $R \in \mathbb{R}^{3 \times 3}$ is the orthogonal rotation matrix of the Gaussian, and $S \in \mathbb{R}^{3 \times 3}$ denotes the diagonal scale matrix of the Gaussian.

With the 3D scene represented as the collection of Gaussians defined in the above way, to render an image given a target viewpoint, inspired by [53], each Gaussian in the collection is first projected onto the image plane corresponding to the viewpoint as:

$$\mu^{2D} = PW\mu, \ \Sigma^{2D} = JW\Sigma W^T J^T \tag{1}$$

where $\mu^{2D}$ and $\Sigma^{2D}$ respectively represent the center and the covariance matrix of the projected Gaussian distribution, $W$ represents the viewing transformation matrix, $P$ represents the projective transformation matrix, and $J$ represents the Jacobian of the affine approximation of the projective transformation. After that, to perform image rendering on the image plane, for each pixel $p$ of the image, its color $C(p)$ is derived through an $\alpha$-blending function as:

$$C(p) = \sum_{i=1}^{N} c_i \beta_i \prod_{j=1}^{i-1} (1 - \beta_j), \ \textbf{where } \beta_i = \alpha_i e^{-\frac{1}{2}(p-\mu_i^{2D})^T (\Sigma_i^{2D})^{-1}(p-\mu_i^{2D}))} \tag{2}$$

where $N$ represents the number of projected Gaussians that overlap $p$, $c_i$ represents the color of the $i$-th Gaussian calculated from its corresponding SH coefficients, $\alpha_i$ represents the opacity of the $i$-th Gaussian, $\mu_i^{2D}$ represents the center of the $i$-th projected Gaussian, and $\Sigma_i^{2D}$ represents the covariance matrix of the $i$-th projected Gaussian. Note that, no matter whether Gaussian Splatting represents the 3D scene using 3D or 2D Gaussian distributions, the above equations can describe its rendering process consistently. In fact, as also mentioned in [26], the difference between rendering images from 3D or 2D Gaussians can be reduced to that, when the scene is represented through 2D Gaussians, the scale matrix $S$ of each of the 2D Gaussians should contain a zero column vector. In this work, we apply our framework to both 2D and 3D Gaussian Splattings, achieving performance improvements as shown in Tab. 2. Yet, as pointed out by [26], using 3D Gaussians instead of 2D Gaussians to represent the scene can result in a viewpoint inconsistency problem. Thus, in Sec. 4, we first focus on explaining how our framework is applied to 2D Gaussian Splatting, in which we fix the last column of the scale matrix $S$ of all the Gaussians to be a zero vector. We then discuss the application of our framework on 3D Gaussian Splatting in Sec. 4.3.

**Cubic Bézier curve.** A cubic Bézier curve is a parametric curve that can be formulated by leveraging a list of four ordered control points $[\omega_0, \omega_1, \omega_2, \omega_3]$ as:

$$B(t) = (1-t)^3 \omega_0 + 3(1-t)^2 t \omega_1 + 3(1-t)t^2 \omega_2 + t^3 \omega_3 \tag{3}$$

In the above equation, we can set $t \in [0, 1]$ for $B(t)$ to represent a segment of the curve that starts from $\omega_0$ and ends at $\omega_3$. Alternatively, we can set $t \in \mathbb{R}$ to represent the entire curve. In this work, we set $t \in \mathbb{R}$ for $B(t)$, as any segment of the curve may not be enough to represent the desired boundaries in the whole image plane. Note that when the four control points lie on the same straight line, the Bézier curve formulated by them would also be reduced to that straight line. Thus, besides representing smooth boundaries, the cubic Bézier curves, at their cross-interacting points, can also be used to represent the sharp corners (of human-made items) in the rendered image.

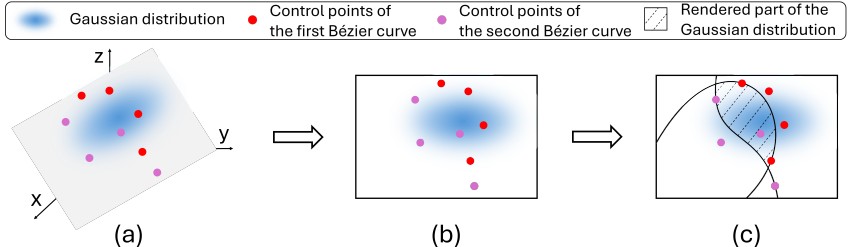

Figure 2: Illustration of the discontinuity-aware rendering process over a single Gaussian distribution. Specifically, over each 2D Gaussian distribution representing the 3D scene, we first introduce it with a new attribute $c_{curve} \in \mathbb{R}^{4M \times 2}$ (represented by the red and purple points in (a)). Here we set $M = 2$. After that, given a viewpoint, as shown in (b), we project both the Gaussian distribution and the points stored in $c_{curve}$ onto the image plane corresponding to the viewpoint. Finally, leveraging the modified $\alpha$-blending function in Eq. 6, we can perform discontinuity-aware rendering and render only the part of the Gaussian distribution masked with the dotted lines in (c). (Best viewed in color.)

## 4 Proposed Method: DisC-GS

Given a batch of source images of a 3D scene with their corresponding viewpoints, the goal of novel view synthesis is to generate novel-view images accurately. To handle this task, a common way is to first learn a 3D scene representation from the given source images. After that, the novel-view images can be rendered from the learned 3D scene. Recently, via representing the 3D scene through Gaussian distributions, Gaussian Splatting has enabled novel-view images to be generated both in real-time and with high rendering quality. It has thus attracted lots of research attention [30, 26, 13, 49].

Yet, we here argue that Gaussian Splatting has a key intrinsic limitation: it may fail to render discontinuities and boundaries accurately. To tackle this problem, in this work, inspired by [54, 28], we propose a novel framework named DisC-GS, which can seamlessly equip Gaussian Spatting with the discontinuity rendering ability. Specifically, during rendering images from the 3D scene, to render discontinuities properly, DisC-GS enables each Gaussian distribution projected onto the image plane to be first "pre-scissored" along certain desired boundaries before being rendered. However, such a "pre-scissoring" operation by itself can break the differentiability of the framework. Considering this, we further incorporate our framework with a Bézier-boundary gradient approximation strategy. Leveraging this strategy, during the learning process of the 3D scene, we can enable the gradient to properly backpropagate through the "pre-scissoring" operation. Below, we first describe the (forward) image rendering process of DisC-GS, and then explain the Bézier-boundary gradient approximation strategy.

### 4.1 Discontinuity-aware Image Rendering

In the proposed DisC-GS, to perform discontinuity-aware rendering, we aim to preprocess Gaussian distributions projected onto the image plane by "scissoring" them along boundaries represented by cubic Bézier curves before rendering. To achieve this, we modify the conventional Gaussian Splatting technique through the following three steps.

**Introduction of an additional attribute.** To facilitate the representation of cubic Bézier curves corresponding to the boundaries of each Gaussian distribution projected on the image plane, we introduce an additional attribute. We denote this attribute $c_{curve} \in \mathbb{R}^{4M \times 2}$, where $M$ is a user-defined hyperparameter representing the number of Bézier curves. This attribute augments the original four attributes (discussed in Sec. 3) of each Gaussian distribution. Below, we introduce the physical interpretation of $c_{curve}$. Specifically, for a certain 2D Gaussian distribution representing the 3D scene, denote the first column of its rotation matrix $R$ to be $r_1$ and the second column of $R$ to be $r_2$. Over the 3D space, the 2D subspace that this Gaussian distribution lies in can be then described by a 2D coordinate system, which takes the center $\mu$ of the Gaussian as its origin, the direction of $r_1$ as the direction of its x-axis, and the direction of $r_2$ as the direction of its y-axis. Then for $c_{curve}$ of this Gaussian distribution, it can be understood as storing a total of $4M$ points in the above-defined coordinate system. Note that when these $4M$ points are projected onto the image plane (as discussed

below), they can then serve as the control points of $M$ cubic Bézier curves, which represent the desired boundary w.r.t. the current Gaussian distribution.

**Image plane projection of points in $c_{curve}$.** After introducing $c_{curve}$ to each Gaussian distribution that represents the 3D scene, given a viewpoint, we project points in $c_{curve}$ onto the image plane. Specifically, this is achieved in two steps: (1) We first transform each point (stored in $c_{curve}$) in the above-defined subspace coordinate system to the coordinate system of the 3D space as:

$$c_{curve}^{3D}[i] = \mu + c_{curve}[i,0] \times r_1^T + c_{curve}[i,1] \times r_2^T, \textbf{ where } i \in \{0, ..., 4M-1\} \tag{4}$$

where $\mu$ is the center of the Gaussian distribution. Note that here, since a column of a rotation matrix is already a unit vector, we can omit the normalization of $r_1$ and $r_2$ and directly transpose them. (2) After deriving $c_{curve}^{3D} \in \mathbb{R}^{4M \times 3}$ storing the $4M$ points in the 3D space coordinate system, we can project each point in $c_{curve}^{3D}$ onto the image plane similar to what we have done in Eq. 1 as:

$$c_{curve}^{2D}[i] = PW c_{curve}^{3D}[i], \textbf{ where } i \in \{0, ..., 4M-1\} \tag{5}$$

where $P$ represents the projective transformation matrix, and $W$ represents the viewing transformation matrix. At this point, for each Gaussian projected onto the image plane via Eq. 1, we have gotten the control points of its desired cubic-Bézier-curves-represented boundary, stored in $c_{curve}^{2D} \in \mathbb{R}^{4M \times 2}$.

**Discontinuity-aware rendering.** Finally, to perform discontinuity-aware image rendering, for each Gaussian distribution projected onto the image plane, we aim to first "scissor" the distribution along the $M$ cubic Bézier curves formulated based on the $4M$ control points stored in $c_{curve}^{2D}$. After that, we would like to only render the remaining parts of the distribution that are not "scissored out". To achieve this, assume that for each projected Gaussian distribution, we have built a binary indicator function $g(\cdot)$, which when passed with a pixel $p$ on the image plane, can output 0 if the pixel is in the "scissored out" area of the distribution, and can output 1 otherwise. We can then perform discontinuity-aware rendering simply via modifying the $\alpha$-blending function in Eq. 2 as:

$$C(p) = \sum_{i=1}^{N} c_i \beta_i \prod_{j=1}^{i-1} (1 - \beta_j), \textbf{ where } \beta_i = \alpha_i g_i(p) e^{-\frac{1}{2}(p - \mu_i^{2D})^T (\Sigma_i^{2D})^{-1} (p - \mu_i^{2D})} \tag{6}$$

where $g_i(\cdot)$ represents the indicator function w.r.t. the $i$-th projected Gaussian. Besides, same as in Eq. 2, $N$ represents the number of projected Gaussians that overlap $p$, $c_i$ represents the color of the $i$-th Gaussian calculated from its corresponding SH coefficients, $\alpha_i$ represents the opacity of the $i$-th Gaussian, $\mu_i^{2D}$ represents the center of the $i$-th projected Gaussian, and $\Sigma_i^{2D}$ represents the covariance matrix of the $i$-th projected Gaussian. Note that via the above modified $\alpha$-blending function, for Gaussians that no longer overlap with the pixel $p$ due to the "scissoring" operation, we can zero out their contributions during calculating the color of $p$.

Considering the above, the problem of enabling Gaussian Splatting to perform discontinuity-aware image rendering has now been reduced to building the indicator function $g(\cdot)$ for each projected Gaussian distribution based on its corresponding $c_{curve}^{2D}$. Below, we discuss how we build $g(\cdot)$. For simplicity, we first consider the case where only one cubic Bézier curve exists per Gaussian distribution. In this case, denote the four control points of the curve $\omega_0 = (x_0, y_0)$, $\omega_1 = (x_1, y_1)$, $\omega_2 = (x_2, y_2)$, and $\omega_3 = (x_3, y_3)$. Then to build $g(\cdot)$, given a pixel $p = (x_p, y_p)$, we just need to determine (judge) whether $p$ is on the inner side or the outer side of the curve. To achieve this, instead of directly leveraging the parametric representation of the cubic Bézier curve presented in Eq. 3, which may lead the judgment to be non-intuitive, we first leverage the implicitization technique in algebra [1] to represent the cubic Bézier curve in its implicit representation form as:

$$B_{imp}(x, y) = \gamma_{xxx} x^3 + \gamma_{xxy} x^2 y + \gamma_{xyy} xy^2 + \gamma_{yyy} y^3 + \gamma_{xx} x^2 + \gamma_{xy} xy + \gamma_{yy} y^2 + \gamma_x x + \gamma_y y + \gamma_0 = 0 \tag{7}$$

where coefficients including $\gamma_{xxx}$, $\gamma_{xxy}$, $\gamma_{xyy}$, $\gamma_{yyy}$, $\gamma_{xx}$, $\gamma_{xy}$, $\gamma_{yy}$, $\gamma_x$, $\gamma_y$, and $\gamma_0$ can all be obtained through basic arithmetic operations over the coordinates of the four control points of the curve in $O(1)$ time complexity (more details are provided in Appendix C). Based on $B_{imp}(x, y)$, in the case where only one curve exists per Gaussian distribution, we can then build the single-curve indicator function $g_{sc}(\cdot)$ intuitively and with $O(1)$ time complexity as:

$$g_{sc}(\omega_0, \omega_1, \omega_2, \omega_3; p) = \begin{cases} 1, & \text{if } B_{imp}(x_p, y_p) > 0, \\ 0, & \text{otherwise} \end{cases} \tag{8}$$

Above we introduce how we can build the indicator function $g(\cdot)$ as $g_{sc}(\cdot)$ assuming that each projected Gaussian distribution is only "scissored" along one cubic Bézier curve. Here, in the case where $M$ curves exist per Gaussian, for each Gaussian, we notice that a pixel can be regarded as in

its "scissored out" area as long as the pixel is "scissored out" by at least one out of the $M$ curves corresponding to the Gaussian. With this in mind, leveraging the $g_{sc}(\cdot)$ function above, we can then define $g(\cdot)$ in cases where $M > 1$ as:

$$g(p) = \prod_{i=0}^{M-1} g_{sc}(c_{curve}^{2D}[4i], c_{curve}^{2D}[4i+1], c_{curve}^{2D}[4i+2], c_{curve}^{2D}[4i+3]; p) \qquad (9)$$

Leveraging the indicator function $g(\cdot)$ defined in Eq. 9, along with the modified $\alpha$-blending function in Eq. 6, we can then enable Gaussian Splatting to perform discontinuity-aware rendering.

## 4.2 Bézier-boundary Gradient Approximation Strategy

Above we discussed, how, in our framework, we perform discontinuity-aware rendering in the forward direction from the 3D scene representation to the 2D rendered image.

**Problems remain.** Yet, this forward rendering process by itself cannot be seamlessly incorporated into the Gaussian Splatting pipeline. This is because of two reasons. Firstly, to enable the 3D scene representation to be properly learned from the source images of the 3D scene, Gaussian Splatting needs its rendering process to be (backward) differentiable. However, performing discontinuity-aware rendering leveraging the modified $\alpha$-blending function in Eq. 6, with the discontinuous function $g(\cdot)$ in Eq. 9 incorporated, is no longer differentiable. Moreover, according to Eq. 8 and 9, $g(\cdot)$ is actually a piecewise constant function. Thus, even in its differentiable segments, the gradients of $g(\cdot)$ w.r.t. its inputs are always zero. In other words, even in segments of $g(\cdot)$ where its gradients are computable, these consistently zero gradients would fail to guide the update of the function $g(\cdot)$'s inputs stored in $c_{curve}^{2D}$, and consequently fail to guide the learning process of $c_{curve}$ introduced in Sec. 4.1.

**The big picture of our proposed strategy.** To tackle the above problems and thus enable Gaussian Splatting to seamlessly render discontinuities, in our framework, we aim to further keep the "differentiability" of the whole discontinuity-aware rendering process. In other words, w.r.t. the discontinuous indicator function $g(\cdot)$ that is newly incorporated into the rendering process, we aim to approximate its gradients (partial derivatives) over the control point coordinates stored in $c_{curve}^{2D}$, in a way that enables the approximated gradients to effectively guide the learning process of the 3D scene representation. To achieve this, inspired by [29], we propose a Bézier-boundary gradient approximation strategy. Below, to ease our explanation of the strategy, we focus on discussing how we approximate $\frac{\partial g}{\partial c_{curve}^{2D}[0,0]}$, i.e., the partial derivative of the indicator function $g(\cdot)$ over the $x$ coordinate of the first control point stored in $c_{curve}^{2D}$. Note that the application of the strategy to the remaining coordinates stored in $c_{curve}^{2D}$ follows a similar process (more details are provided in Appendix D). Specifically, to approximate $\frac{\partial g}{\partial c_{curve}^{2D}[0,0]}$, based on the chain rule and according to Eq. 9, denoting $g_{sc}(c_{curve}^{2D}[0], c_{curve}^{2D}[1], c_{curve}^{2D}[2], c_{curve}^{2D}[3]; p)$ to be $g_{sc}^0(p)$, we first have:

$$\frac{\partial g}{\partial c_{curve}^{2D}[0,0]} = \frac{\partial g}{\partial g_{sc}^0(p)} \times \frac{\partial g_{sc}^0(p)}{\partial c_{curve}^{2D}[0,0]} \qquad (10)$$

Then since $g(\cdot)$ is clearly differentiable over $g_{sc}^0(\cdot)$ based on its definition in Eq. 9, we can reduce our problem to approximate $\frac{\partial g_{sc}^0(p)}{\partial c_{curve}^{2D}[0,0]}$, which is achieved through the following two steps.

**Determining if $g_{sc}^0(p)$ is desired to be modified.** Specifically, to approximate $\frac{\partial g_{sc}^0(p)}{\partial c_{curve}^{2D}[0,0]}$, we first would like to determine, if the function $g_{sc}^0(\cdot)$ is desired to be modified at $p$ or not. This is because, if $g_{sc}^0(\cdot)$ already outputs a satisfied value at $p$, we don't need to change $c_{curve}^{2D}[0,0]$ to correspondingly modify $g_{sc}^0(p)$. In other words, in such a case, we can simply set $\frac{\partial g_{sc}^0(p)}{\partial c_{curve}^{2D}[0,0]}$ to be zero.

Denote the loss function used during the learning process to be $L$. Leveraging both $\frac{\partial L}{\partial g_{sc}^0(p)}$ and the current value of $g_{sc}^0(p)$ as the conditions, below, we list the three situations in which $g_{sc}^0(p)$ doesn't need to be further modified: (1) The first situation happens when $\frac{\partial L}{\partial g_{sc}^0(p)} = 0$, which indicates that $g_{sc}^0(\cdot)$ given input $p$ is already in an optimal state. (2) Besides, the second situation happens when $\frac{\partial L}{\partial g_{sc}^0(p)} > 0$ and $g_{sc}^0(p) = 0$. Based on the gradient descent algorithm, this implies that, while we still hope the function $g_{sc}^0(\cdot)$ to output a smaller value at $p$, the function $g_{sc}^0(\cdot)$ already outputs its smallest allowed value. (3) Following the opposite logic of situation (2), the third situation happens when

$\frac{\partial L}{\partial g_{sc}^0(p)} < 0$ and $g_{sc}^0(p) = 1$. In this case, though we still want $g_{sc}^0(p)$ to be larger, $g_{sc}^0(\cdot)$ at $p$ already outputs its largest allowed value. In the above three situations, we can directly set $\frac{\partial g_{sc}^0(p)}{\partial c_{curve}^{2D}[0,0]} = 0$ and omit the approximation performed in the next step.

**Approximating** $\frac{\partial g_{sc}^0(p)}{\partial c_{curve}^{2D}[0,0]}$. Besides the above three situations, in the rest cases, for the value of function $g_{sc}^0(\cdot)$ at $p$ to be properly modified based on the modification of the value of $c_{curve}^{2D}[0,0]$, we aim to properly approximate the partial derivative $\frac{\partial g_{sc}^0(p)}{\partial c_{curve}^{2D}[0,0]}$. To achieve this, recall that as a binary indicator function, $g_{sc}^0(\cdot)$ switches (modifies) its value at $p$ between 0 and 1 only when its corresponding cubic Bézier curve passes through $p$. Considering this, below, we first identify: which value we should set (change) $c_{curve}^{2D}[0,0]$ to be, so that the value switch of $g_{sc}^0(\cdot)$ at $p$ can happen.

To achieve this identification in an intuitive and analytical way, denoting $p = (x_p, y_p)$ and the desired value of $c_{curve}^{2D}[0,0]$ to be $\phi$, based on Eq. 3, we can first derive the following system of equations:

$$\begin{cases} x_p = (1-t)^3\phi + 3(1-t)^2t(c_{curve}^{2D}[1,0]) + 3(1-t)t^2(c_{curve}^{2D}[2,0]) + t^3(c_{curve}^{2D}[3,0]) \\ y_p = (1-t)^3(c_{curve}^{2D}[0,1]) + 3(1-t)^2t(c_{curve}^{2D}[1,1]) + 3(1-t)t^2(c_{curve}^{2D}[2,1]) + t^3(c_{curve}^{2D}[3,1]) \end{cases}$$
(11)

In this system of equations, since $x_p$, $y_p$, and the coordinates in $c_{curve}^{2D}$ all have known values, we initially regard the second equation in the system as a cubic equation w.r.t. $t$, as $t$ is now the only unknown variable in this equation. After solving this cubic equation and with $t$ also known, we can then regard the first equation in the system as a cubic equation w.r.t. $\phi$ and solve it. Finally, by solving the above two equations (both in just $O(1)$ time complexity), we obtain $S_\phi$ as the set of all possible real number solutions for $\phi$. Based on the solutions' scenarios within $S_\phi$, we approximate $\frac{\partial g_{sc}^0(p)}{\partial c_{curve}^{2D}[0,0]}$ in three different ways below.

(1) The "no side" situation. The first situation happens when $S_\phi = \varnothing$. In this case, we simply set $\frac{\partial g_{sc}^0(p)}{\partial c_{curve}^{2D}[0,0]} = 0$. This is because, the empty nature of $S_\phi$ implies that, there exists no proper real-number value that we can change $c_{curve}^{2D}[0,0]$ to be, such that $g_{sc}^0(p)$ can be desirably modified (i.e., either from 0 to 1 or from 1 to 0). We thus simply do not encourage $c_{curve}^{2D}[0,0]$ to change.

(2) The "single side" situation. The second situation occurs when all solutions in $S_\phi$ lie on the same side of $c_{curve}^{2D}[0,0]$ (i.e., all larger or all smaller than $c_{curve}^{2D}[0,0]$). In this situation, let $\widetilde{\phi}$ denote the solution in $S_\phi$ that is nearest to $c_{curve}^{2D}[0,0]$. Adjusting $c_{curve}^{2D}[0,0]$ towards $\widetilde{\phi}$ then implies the least-cost plan, facilitating the modification of $g_{sc}^0(p)$ in a desired manner. With this in mind, to encourage $c_{curve}^{2D}[0,0]$ to approach $\widetilde{\phi}$, inspired by previous studies [6, 10, 47, 29], we approximate $\frac{\partial g_{sc}^0(p)}{\partial c_{curve}^{2D}[0,0]}$ via performing linear interpolation between $c_{curve}^{2D}[0,0]$ and $\widetilde{\phi}$ as:

$$\frac{\partial g_{sc}^0(p)}{\partial c_{curve}^{2D}[0,0]} = \frac{\widetilde{g_{sc}^0(p)} - g_{sc}^0(p)}{\left(\widetilde{\phi} - (c_{curve}^{2D}[0,0])\right) + \epsilon}, \textbf{ where } \widetilde{g_{sc}^0(p)} = \begin{cases} 1, & \text{if } g_{sc}^0(p) = 0, \\ 0, & \text{otherwise} \end{cases}$$
(12)

In the above equation, we set $\epsilon = 10^{-5}$ if $\left(\widetilde{\phi} - (c_{curve}^{2D}[0,0])\right) > 0$ and we otherwise set $\epsilon = -10^{-5}$. $\epsilon$ here is a small number that is used to avoid the gradient exploding problem to happen when the distance between $\widetilde{\phi}$ and $c_{curve}^{2D}[0,0]$ is too short.

(3) The "both sides" situation. The third situation happens when some solutions in $S_\phi$ are on the left side of $c_{curve}^{2D}[0,0]$, while other solutions are on the right side of $c_{curve}^{2D}[0,0]$. In this situation, we can achieve the desired modification of $g_{sc}^0(p)$ via either moving $c_{curve}^{2D}[0,0]$ to its left or right side. Thus, unlike the scenario described in the above situation (2) where we only consider $\widetilde{\phi}$ from a single side of $c_{curve}^{2D}[0,0]$, here, denoting $\widetilde{\phi_1}$ as the value that is nearest to $c_{curve}^{2D}[0,0]$ from its left side, and $\widetilde{\phi_2}$ as the value that is nearest to $c_{curve}^{2D}[0,0]$ from its right side, we approximate $\frac{\partial g_{sc}^0(p)}{\partial c_{curve}^{2D}[0,0]}$ as:

$$\frac{\partial g_{sc}^0(p)}{\partial c_{curve}^{2D}[0,0]} = \frac{\widetilde{g_{sc}^0(p)} - g_{sc}^0(p)}{\left(\widetilde{\phi_1} - (c_{curve}^{2D}[0,0])\right) + \epsilon_1} + \frac{\widetilde{g_{sc}^0(p)} - g_{sc}^0(p)}{\left(\widetilde{\phi_2} - (c_{curve}^{2D}[0,0])\right) + \epsilon_2}$$
(13)

In the above equation, we define $\widetilde{g_{sc}^0(p)}$ in the same way as in Eq. 12. Besides, both $\epsilon_1$ and $\epsilon_2$ are defined in the similar way as $\epsilon$ in Eq. 12.

In summary, taking $\frac{\partial g}{\partial c_{curve}^{2D}[0,0]}$ as an example, the above discussion explains how our proposed strategy approximates the gradient of $g(\cdot)$ with respect to the point coordinates stored in $c_{curve}^{2D}$. With the incorporation of this strategy into our framework, we keep the "differentiability" of the whole rendering process, allowing Gaussian Splatting to seamlessly perform discontinuity-aware rendering.

### 4.3 DisC-GS on 3D Gaussian Splatting

Above, we focus on describing how we use 2D Bézier curves in our DisC-GS framework and correspondingly apply DisC-GS on 2D Gaussian Splatting. Here in this subsection, we further describe how we use 3D Bézier curves in our DisC-GS framework and apply DisC-GS on 3D Gaussian Splatting. Specifically, the transition from 2D to 3D Bézier curves in DisC-GS requires only two minimal modifications. (1) Firstly, for each Gaussian representing the 3D scene, the control points of its Bézier curves are stored directly in the 3D spatial coordinate system rather than in a 2D subspace. Note that, this modification can be very simply made. Specifically, for each Gaussian in the 3D space in our DisC-GS framework, we only need to use $c_{curve}^{3D} \in \mathbb{R}^{4M \times 3}$ instead of $c_{curve} \in \mathbb{R}^{4M \times 2}$ to represent the control point coordinates of its 3D Bézier curves. In other words, for each 3D Gaussian, we only need to introduce it with $c_{curve}^{3D}$ instead of $c_{curve}$ as its new attribute. (2) Moreover, since we already directly introduce $c_{curve}^{3D}$ as the new attribute for each 3D Gaussian in our framework, during rendering, we omit Eq. 4 above in Sec. 4.1, which originally is used to acquire $c_{curve}^{3D}$ from $c_{curve}$. Overall, the above two modifications are sufficient to incorporate DisC-GS with 3D instead of 2D Bézier curves.

### 4.4 Overall Training and Testing

In DisC-GS, during training (i.e., learning the 3D scene representation from the source images), we follow a similar process as the typical Gaussian Splatting technique [30]. The involvement of the strategy introduced in Sec. 4.2 keeps the "differentiability" of our framework. During testing (i.e., image rendering), we use the discontinuity-aware image rendering process introduced in Sec. 4.1.

## 5  Experiments

**Datasets.** To evaluate the efficacy of our proposed framework DisC-GS, following previous Gaussian Splatting works [30, 49], we evaluate our framework on a total of 13 3D scenes, which include both outdoor scenes and indoor scenes. Specifically, among these 13 scenes, 9 of them are from the Mip-NeRF360 dataset [4], 2 of them are from the Tanks&Temples dataset [31], and 2 of them are from the Deep Blending dataset [23]. We also follow previous works [30, 49] in their train-test-split.

**Evaluation metrics.** Following [30, 49], we use the following three metrics for evaluation: Peak Signal-to-Noise Ratio (PSNR), Structural Similarity Index Measure (SSIM), and Learned Perceptual Image Patch Similarity (LPIPS) [51].

**Implementation details.** We conduct our experiments on an RTX 3090 GPU and develop our code mainly based on the GitHub repository [2] provided by Kerbl et al [30]. Moreover, we also get inspired by [35, 52, 46] during our code implementation, and make use of the LPIPS loss during our training process. Furthermore, for the newly introduced attribute $c_{curve} \in \mathbb{R}^{4M \times 2}$, we set its initial learning rate to 2e-4, and set the hyperparameter $M$ to 3. Besides, in the densification procedure of our framework, when a Gaussian is cloned/splitted into two new Gaussians, we assign both the new Gaussians with the same attribute $c_{curve}$ as the original one.

### 5.1  Experimental Results

In Tab. 1, we compare our approach (applied on 2D Gaussian Splatting) with existing novel view synthesis methods evaluated on the same 13 3D scenes and report the PSNR, SSIM, and LPIPS results. Our framework consistently outperforms other methods on all three metrics and across various datasets, showing its effectiveness. We also show qualitative results in both Fig. 1(c) and Appendix B. As shown, whether representing the 3D scene through 3D or 2D Gaussian distributions,

Table 1: Performance comparison on the Tanks&Temples, Mip-NeRF360, and Deep Blending datasets.

| Method | Tanks&Temples | | | Mip-NeRF360 Dataset | | | Deep Blending | | |
|---|---|---|---|---|---|---|---|---|---|
| | SSIM↑ | PSNR↑ | LPIPS↓ | SSIM↑ | PSNR↑ | LPIPS↓ | SSIM↑ | PSNR↑ | LPIPS↓ |
| Plenoxels [18] | 0.719 | 21.08 | 0.379 | 0.626 | 23.08 | 0.463 | 0.795 | 23.06 | 0.510 |
| INGP-Base [37] | 0.723 | 21.72 | 0.330 | 0.671 | 25.30 | 0.371 | 0.797 | 23.62 | 0.423 |
| INGP-Big [37] | 0.745 | 21.92 | 0.305 | 0.699 | 25.59 | 0.331 | 0.817 | 24.96 | 0.390 |
| Mip-NeRF360 [4] | 0.759 | 22.22 | 0.257 | 0.792 | 27.69 | 0.237 | 0.901 | 29.40 | 0.245 |
| 3D-GS [30] | 0.841 | 23.14 | 0.183 | 0.815 | 27.21 | 0.214 | 0.903 | 29.41 | 0.243 |
| Surfsplatting [26] | 0.837 | 23.42 | 0.202 | 0.804 | 27.03 | 0.239 | 0.895 | 28.89 | 0.261 |
| FreGS [49] | 0.849 | 23.96 | 0.178 | 0.826 | 27.85 | 0.209 | 0.904 | 29.93 | 0.240 |
| GES [22] | 0.836 | 23.35 | 0.198 | 0.794 | 26.91 | 0.250 | 0.901 | 29.68 | 0.252 |
| Mip-Splatting [48] | 0.851 | 23.78 | 0.178 | 0.827 | 27.79 | 0.203 | 0.904 | 29.69 | 0.248 |
| Ours | **0.866** | **24.96** | **0.120** | **0.833** | **28.01** | **0.189** | **0.907** | **30.42** | **0.199** |

the conventional Gaussian Splatting technique often struggles to render boundaries and discontinuities clearly and with high quality. In contrast, our framework can achieve good rendering quality, even in regions of the image containing numerous boundaries and discontinuities. This further underscores the efficacy of our approach.

## 5.2 Ablation Studies

We conduct extensive ablation experiments on the Tanks&Temples dataset. **More ablation studies w.r.t. the image areas with rich boundaries, the Bézier-boundary gradient approximation strategy, the hyperparameters, and the rendering speed of our framework are in Appendix A.**

**Impact of representing the scene with 2D or 3D Gaussians in DisC-GS.** In Sec. 4.1 and Sec. 3, we focus on introducing how we apply DisC-GS on 2D Gaussian Splatting. After that, in Sec. 4.3, we introduce how DisC-GS can be applied on 3D Gaussian Splatting in a similar way. Here to verify the generality of our framework, we test applying our framework on both 2D and 3D Gaussian Splatting. As shown in Tab. 2, our framework, when applied on both 2D and 3D Gaussian Splattings, can consistently achieve performance improvements, demonstrating the generality of our framework.

Table 2: Evaluation of our framework on both 2D and 3D Gaussian Splattings.

| Method | SSIM↑ | PSNR↑ | LPIPS↓ |
|---|---|---|---|
| 2D Gaussian Splatting | 0.836 | 23.30 | 0.205 |
| 2D Gaussian Splatting + Ours | 0.866 | 24.96 | 0.120 |
| 3D Gaussian Splatting | 0.841 | 23.14 | 0.183 |
| 3D Gaussian Splatting + Ours | 0.863 | 24.67 | 0.123 |

**Impact of the number of control points per Bézier curve.** In our framework, inspired by [16, 34], we represent boundaries in the image with the cubic Bézier curve, each of which is formulated by leveraging 4 control points. Here we evaluate formulating each Bézier curve by other numbers of control points, and report the

Table 3: Evaluation on the number of control points per Bézier curve.

| Method | SSIM↑ | PSNR↑ | LPIPS↓ |
|---|---|---|---|
| 2 control points per curve | 0.853 | 24.14 | 0.138 |
| 3 control points per curve | 0.861 | 24.58 | 0.127 |
| 4 control points per curve | 0.866 | 24.96 | 0.120 |
| 5 control points per curve | 0.863 | 24.68 | 0.126 |

results in Tab. 3. As shown, our framework gets optimal performance when the number of control points per Bézier curve is set to 4, and we thus formulate each Bézier curve by utilizing 4 control points in our experiments. Besides, with different choices of the number of control points per Bézier curve from 2 to 5, our framework outperforms the previous state-of-the-art method consistently. This shows the robustness of our framework to the number of control points per Bézier curve.

## 6 Conclusion

In this paper, we have proposed an innovative novel view synthesis framework DisC-GS, which for the first time, enables Gaussian Splatting to properly represent and render discontinuities and boundaries in its image rendering process. Moreover, to keep the "differentiability" of our framework, we further introduce our framework with a Bézier-boundary gradient approximation strategy. Our framework consistently achieves superior performance across different evaluation benchmarks.

**Limitations.** While our framework enables Gaussian Splatting to perform discontinuity-aware rendering, we acknowledge that same as existing Gaussian Splatting approaches, our framework still holds certain limitations, such as challenges in rendering large scenes.

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

# A  Additional Ablation Studies

In this section, we conduct more ablation experiments on the Tanks&Temples dataset.

**Evaluation especially over areas with rich boundaries in the testing images.** In this work, we point out that Gaussian Splatting holds an

Table 4: Evaluation especially over areas with rich boundaries in the testing images.

| Method | MaskedSSIM↑ | |
|---|---|---|
| | Boundary-rich areas | Boundary-sparse areas |
| Baseline(2D Gaussian Splatting) | 0.819 | 0.922 |
| Ours | 0.855 | 0.934 |

inherent weakness in rendering discontinuities and boundaries, and we propose a DisC-GS framework for Gaussian Splatting to render boundaries in the image more accurately. Here, we aim to test our framework, particularly over image areas with rich boundaries. To achieve this, we evaluate our framework respectively over two parts of areas in each testing image. Specifically, for each testing image, we first pass the image over the Canny algorithm [9] followed by the dilation operation to highlight the areas with rich boundaries in the image (i.e., the areas that involve or surround the Canny-detected boundaries). After that, we let the first part (**boundary-rich areas**) include all the highlighted areas in each testing image, and let the second part (**boundary-sparse areas**) include all the rest areas in each testing image. To perform evaluation effectively over a part rather than the whole of each testing image, following [41], we use MaskedSSIM as the evaluation metric. As shown in Tab. 4, compared to 2D Gaussian Splatting as a baseline, especially in the boundary-rich areas, our framework can achieve a significant performance improvement. This demonstrates the effectiveness of our method especially over image areas with rich boundaries.

**Impact of introducing different Gaussians with different numbers of Bézier curves.** In our framework, for each Gaus-

Table 5: Evaluation on introducing different Gaussians with different numbers of Bézier curves.

| Method | SSIM↑ | PSNR↑ | LPIPS↓ |
|---|---|---|---|
| Different numbers of curves for different Gaussians | 0.863 | 24.79 | 0.122 |
| $M$ curves for each Gaussian | 0.866 | 24.96 | 0.120 |

sian distribution representing the 3D scene, we introduce it with $M$ cubic Bézier curves ($M$ **curves for each Gaussian**). Here, to investigate whether introducing different Gaussians with different numbers of curves can further benefit our framework, we test another variant (**different numbers of curves for different Gaussians**). Specifically, in this variant, before the start of the training process w.r.t. a certain 3D scene, for each of the source images, we first identify its boundary-rich areas similarly as in the above ablation study. After that, during training, when a Gaussian distribution is newly created through the adaptive control process of Gaussian Splatting, instead of directly introducing it with $M$ curves, we introduce the Gaussian with $M + 1$ Bézier curves if the center $\mu^{2D}$ of its corresponding projected Gaussian lies in the highlighted boundary-rich areas. Otherwise, we introduce the newly created Gaussian with $M - 1$ Bézier curves. As shown in Tab. 5, this variant does not result in better performance compared to our framework. This might be because, during training, for each Gaussian representing the 3D scene, its center is learnable, and the Gaussian is thus movable. In other words, for Gaussians that are initialized with more curves and thus may be able to represent boundaries more accurately, they can move to areas with fewer (or no) boundaries in the 3D scene. At the same time, for Gaussians that are initialized with fewer curves, they can also move to areas with rich boundaries in the 3D scene. With the above in mind, in our experiments, we introduce each Gaussian representing the 3D scene consistently with the same number of $M$ Bézier curves, equipping each Gaussian with the same level of power in boundary representation.

**Image sharpness evaluation.** In this work, we propose DisC-GS, which enables Gaussian Splatting to render the sharp boundaries in the image more accurately. Considering this, to perform further evaluation of our framework, we here also evaluate our framework from the im-

Table 6: Evaluation on the sharpness of the rendered images.

| Method | Image sharpness |
|---|---|
| Baseline(2D Gaussian Splatting) | 51.50 % |
| Ours | 57.72 % |

age sharpness perspective. Specifically, following [15], we measure image sharpness leveraging the energy gradient function. As shown in Tab. 6, with our framework applied, we can render images more sharply. This further shows the efficacy of our proposed framework.

**Impact of the "both sides" situation.** To keep the "differentiability" of our framework, in this work, we propose a Bézier-boundary gradient approximation strategy. Specifically, in this

Table 7: Evaluation on the "both sides" situation.

| Method | SSIM↑ | PSNR↑ | LPIPS↓ |
|---|---|---|---|
| w/o the "both sides" situation | 0.854 | 24.38 | 0.135 |
| with the "both sides" situation | 0.866 | 24.96 | 0.120 |

strategy, when some solutions in $S_\phi$ lie on the left side of $c_{curve}^{2D}[0,0]$ while other solutions lie on its right side, we approximate $\frac{\partial g_{sc}^0(p)}{\partial c_{curve}^{2D}[0,0]}$ through Eq. 13 under the "both sides" situation, considering both the left and right sides of $c_{curve}^{2D}[0,0]$ (**with the "both sides" situation**). To valid this design, we test a variant (**w/o the "both sides" situation**), even when solutions in $S_\phi$ exist on both the left and right sides of $c_{curve}^{2D}[0,0]$, we still consider only the solution that is nearest to $c_{curve}^{2D}[0,0]$, and approximate $\frac{\partial g_{sc}^0(p)}{\partial c_{curve}^{2D}[0,0]}$ through Eq. 12 under the "single side" situation. As shown in Tab. 7, our framework outperforms this variant. This shows the advantage of considering "both sides" when solutions in $S_\phi$ lie on both the left and right sides of $c_{curve}^{2D}[0,0]$.

**Impact of the small numbers $\epsilon$, $\epsilon_1$, and $\epsilon_2$.** In our framework, during gradient approximation, to avoid the gradient exploding problem to happen, we add $\epsilon$ as a small number to the denominator part of Eq. 12, and we also add $\epsilon_1$ and $\epsilon_2$ to Eq. 13 in a similar way (**with small numbers**). To valid the efficacy of this

Table 8: Evaluation on the small numbers $\epsilon$, $\epsilon_1$, and $\epsilon_2$.

| Method | SSIM↑ | PSNR↑ | LPIPS↓ |
|---|---|---|---|
| w/o small numbers | 0.858 | 24.37 | 0.130 |
| with small numbers | 0.866 | 24.96 | 0.120 |

design, we test a variant (**w/o small numbers**) in which we remove $\epsilon$, $\epsilon_1$, and $\epsilon_2$ from our gradient approximation process. As shown in Tab. 8, our framework involving these small numbers performs better than this variant, showing the efficacy of these small numbers.

**Impact of the number of Bézier curves per Gaussian $M$.** In our framework, for each Gaussian distribution representing the 3D scene, we set the number of cubic Bézier curves $M$ associate with the Gaussian to 3. As shown in Tab. 9, our framework achieves optimal performance when $M$ is set to 3, and $M = 3$ is used in our experiments. Moreover, with different choices of $M$ from 1 to 4, our framework consistently achieves good performance. This

Table 9: Evaluation on the number of Bézier curves per Gaussian $M$.

| Method | SSIM↑ | PSNR↑ | LPIPS↓ |
|---|---|---|---|
| $M = 1$ | 0.853 | 24.36 | 0.139 |
| $M = 2$ | 0.863 | 24.76 | 0.124 |
| $M = 3$ | 0.866 | 24.96 | 0.120 |
| $M = 4$ | 0.862 | 24.81 | 0.125 |

demonstrates the robustness of our proposed framework to this hyperparameter.

**Impact of the initial learning rate set to $c_{curve}$.** In our framework, we introduce a new attribute $c_{curve}$, for which in our experiments, we set its initial learning rate ($lr_{curve}$) to 2e-4. Here we also assess the other choices of $lr_{curve}$ from 1e-4 to 1e-3 and report the results in Tab. 10. As shown, with different choices of $lr_{curve}$, the performance of our framework is consistent, which shows the robustness of our framework to $lr_{curve}$.

Table 10: Evaluation on the initial learning rate ($lr_{curve}$) set to $c_{curve}$.

| Method | SSIM↑ | PSNR↑ | LPIPS↓ |
|---|---|---|---|
| $lr_{curve} = 1e-4$ | 0.861 | 24.60 | 0.126 |
| $lr_{curve} = 2e-4$ | 0.866 | 24.96 | 0.120 |
| $lr_{curve} = 5e-4$ | 0.861 | 24.64 | 0.125 |
| $lr_{curve} = 1e-3$ | 0.857 | 24.30 | 0.132 |

**Rendering time.** In Tab. 11, we compare the rendering time of our framework with the existing NeRF-based method Mip-NeRF360 [4], as well as two Gaussian Splatting baselines (i.e., 2D Gaussian Splatting and 3D Gaussian Splatting), on an RTX 3090 GPU in terms of seconds per image. As shown, our DisC-GS can achieve a competitive rendering time (speed) compared to the existing methods leveraging

Table 11: Analysis of rendering time in terms of seconds per image. Our framework can run efficiently and satisfy most real-time requirements, yet achieves superior performance.

| Method | PSNR↑ | Rendering time |
|---|---|---|
| Mip-NeRF360 [4] | 22.22 | 7.143s |
| 2D Gaussian Splatting | 23.30 | 0.007s |
| 3D Gaussian Splatting | 23.14 | 0.007s |
| Ours (on 2D Gaussian Splatting) | 24.96 | 0.008s |
| Ours (on 3D Gaussian Splatting) | 24.67 | 0.008s |

the conventional Gaussian Splatting technique, while obtaining much better performance.

| 2D Gaussian Splatting w/o DisC-GS | 2D Gaussian Splatting with DisC-GS | Ground Truth |
|---|---|---|

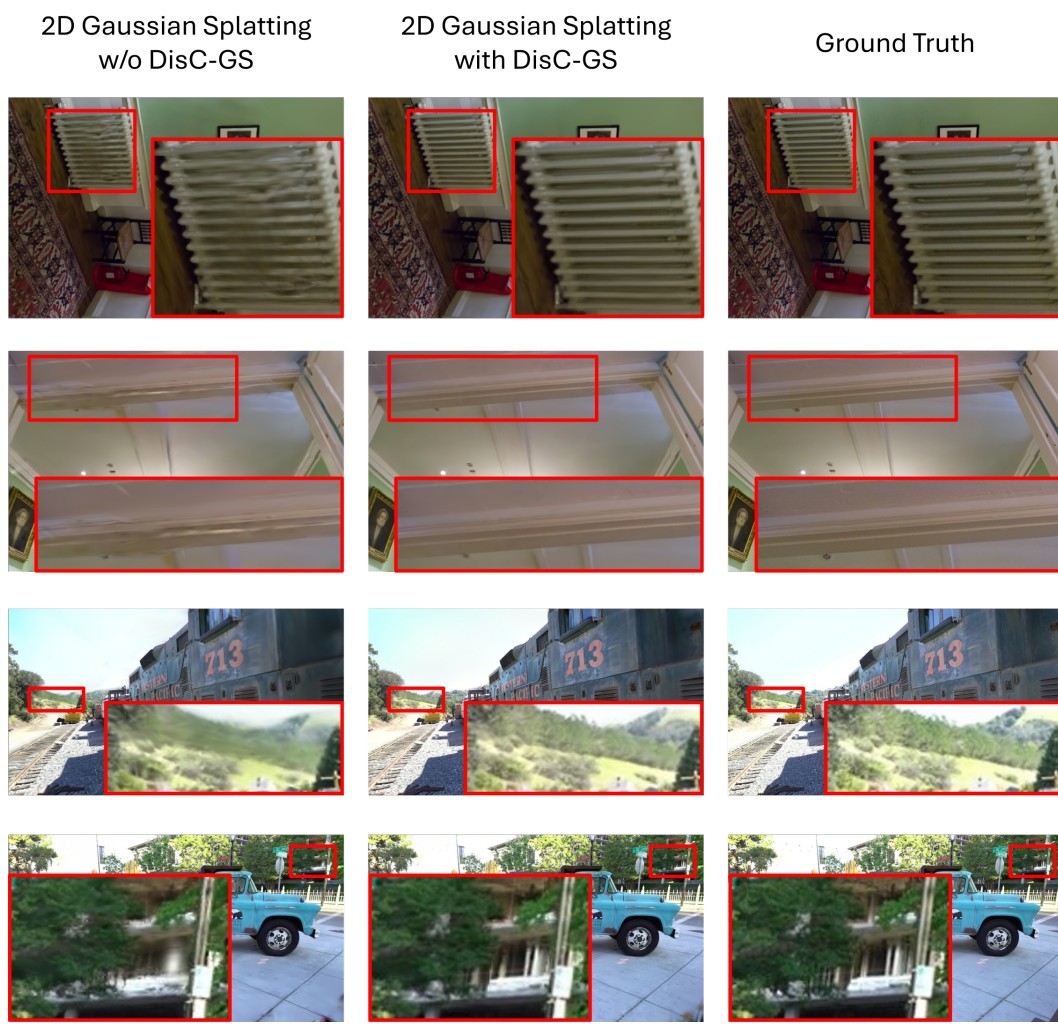

Figure 3: Qualitative results of 2D Gaussian Splatting with and without DisC-GS.

## B  Additional Qualitative Results

In this section, we present more qualitative results. Specifically, in Fig. 3, we present images rendered by 2D Gaussian Splatting with and without applying our proposed framework DisC-GS; in Fig. 4, we present images rendered by 3D Gaussian Splatting with and without DisC-GS. As shown, no matter representing the 3D scene through 2D or 3D Gaussian distributions, the typical Gaussian Splatting technique can fail to render boundaries and discontinuities clearly and with high quality. Yet, our DisC-GS framework, when applied, can achieve good rendering quality, even in regions of the image containing numerous boundaries and discontinuities. This further shows the efficacy of our approach.

## C  Additional Details about Eq. 7 in the Main Paper

In Eq. 7 in the main paper, via leveraging the implicitization technique in algebra [1], we enable the cubic Bézier curve to be represented in its implicit representation form as:

$$B_{imp}(x, y) = \gamma_{xxx}x^3 + \gamma_{xxy}x^2y + \gamma_{xyy}xy^2 + \gamma_{yyy}y^3 + \gamma_{xx}x^2 + \gamma_{xy}xy + \gamma_{yy}y^2 + \gamma_x x + \gamma_y y + \gamma_0 = 0 \tag{14}$$

Here in this section, we discuss how we derive coefficients including $\gamma_{xxx}$, $\gamma_{xxy}$, $\gamma_{xyy}$, $\gamma_{yyy}$, $\gamma_{xx}$, $\gamma_{xy}$, $\gamma_{yy}$, $\gamma_x$, $\gamma_y$, and $\gamma_0$ in Eq. 7 in the main paper (re-shown in Eq. 14 above).

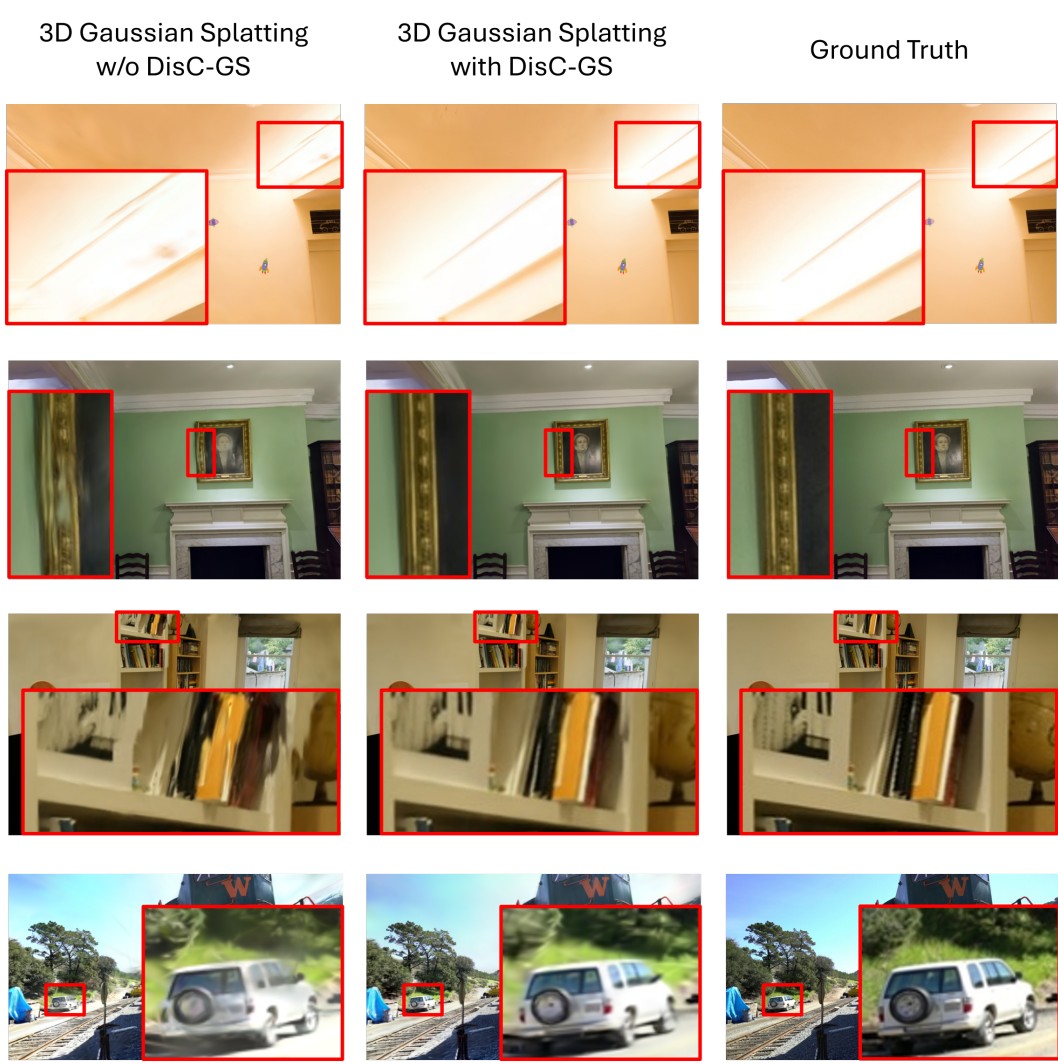

Figure 4: Qualitative results of 3D Gaussian Splatting with and without DisC-GS.

Specifically, denote the four control points of the cubic Bézier curve $\omega_0 = (x_0, y_0)$, $\omega_1 = (x_1, y_1)$, $\omega_2 = (x_2, y_2)$, and $\omega_3 = (x_3, y_3)$. To derive the coefficients in Eq. 14, following [1], we first define a set of intermediate coefficients as:

$$
\begin{aligned}
\zeta_0 &= x_0; \\
\zeta_1 &= -3 \times x_0 + 3 \times x_1; \\
\zeta_2 &= -6 \times x_1 + 3 \times x_0 + 3 \times x_2; \\
\zeta_3 &= x_3 - x_0 - 3 \times x_2 + 3 \times x_1; \\
\zeta_4 &= y_0; \\
\zeta_5 &= -3 \times y_0 + 3 \times y_1; \\
\zeta_6 &= -6 \times y_1 + 3 \times y_0 + 3 \times y_2; \\
\zeta_7 &= y_3 - y_0 - 3 \times y_2 + 3 \times y_1;
\end{aligned}
$$

After that, utilizing these intermediate coefficients, following [1], we can then compute coefficients in Eq. 14 as:

$\gamma_{xxx} = \zeta_7 \times \zeta_7 \times \zeta_7;$

$\gamma_{xxy} = -3 \times \zeta_3 \times \zeta_7 \times \zeta_7;$

$\gamma_{xyy} = 3 \times \zeta_7 \times \zeta_3 \times \zeta_3;$

$\gamma_{yyy} = -\zeta_3 \times \zeta_3 \times \zeta_3;$

$$\gamma_{xx} = -3 \times \zeta_3 \times \zeta_5 \times \zeta_6 \times \zeta_7 + \zeta_1 \times \zeta_6 \times \zeta_7 \times \zeta_7 - \zeta_2 \times \zeta_7 \times \zeta_6 \times \zeta_6 + 2 \times \zeta_2 \times \zeta_5 \times \zeta_7 \times \zeta_7$$
$$+ 3 \times \zeta_3 \times \zeta_4 \times \zeta_7 \times \zeta_7 + \zeta_3 \times \zeta_6 \times \zeta_6 \times \zeta_6 - 3 \times \zeta_0 \times \zeta_7 \times \zeta_7 \times \zeta_7;$$

$$\gamma_{xy} = \zeta_1 \times \zeta_3 \times \zeta_6 \times \zeta_7 - \zeta_2 \times \zeta_3 \times \zeta_5 \times \zeta_7 - 6 \times \zeta_4 \times \zeta_7 \times \zeta_3 \times \zeta_3 - 3 \times \zeta_1 \times \zeta_2 \times \zeta_7 \times \zeta_7$$
$$- 2 \times \zeta_2 \times \zeta_3 \times \zeta_6 \times \zeta_6 + 2 \times \zeta_6 \times \zeta_7 \times \zeta_2 \times \zeta_2 + 3 \times \zeta_5 \times \zeta_6 \times \zeta_3 \times \zeta_3$$
$$+ 6 \times \zeta_0 \times \zeta_3 \times \zeta_7 \times \zeta_7;$$

$$\gamma_{yy} = 3 \times \zeta_1 \times \zeta_2 \times \zeta_3 \times \zeta_7 + \zeta_3 \times \zeta_6 \times \zeta_2 \times \zeta_2 - \zeta_2 \times \zeta_5 \times \zeta_3 \times \zeta_3 - 3 \times \zeta_0 \times \zeta_7 \times \zeta_3 \times \zeta_3$$
$$- 2 \times \zeta_1 \times \zeta_6 \times \zeta_3 \times \zeta_3 - \zeta_7 \times \zeta_2 \times \zeta_2 \times \zeta_2 + 3 \times \zeta_4 \times \zeta_3 \times \zeta_3 \times \zeta_3;$$

$$\gamma_x = \zeta_2 \times \zeta_3 \times \zeta_4 \times \zeta_5 \times \zeta_7 - \zeta_1 \times \zeta_2 \times \zeta_5 \times \zeta_6 \times \zeta_7 - \zeta_1 \times \zeta_3 \times \zeta_4 \times \zeta_6 \times \zeta_7$$
$$+ 6 \times \zeta_0 \times \zeta_3 \times \zeta_5 \times \zeta_6 \times \zeta_7 + \zeta_5 \times \zeta_1 \times \zeta_1 \times \zeta_7 \times \zeta_7 + \zeta_7 \times \zeta_2 \times \zeta_2 \times \zeta_5 \times \zeta_5$$
$$+ 3 \times \zeta_7 \times \zeta_3 \times \zeta_3 \times \zeta_4 \times \zeta_4 + \zeta_1 \times \zeta_3 \times \zeta_5 \times \zeta_6 \times \zeta_6 - \zeta_2 \times \zeta_3 \times \zeta_6 \times \zeta_5 \times \zeta_5$$
$$- 6 \times \zeta_0 \times \zeta_3 \times \zeta_4 \times \zeta_7 \times \zeta_7 - 4 \times \zeta_0 \times \zeta_2 \times \zeta_5 \times \zeta_7 \times \zeta_7 - 3 \times \zeta_4 \times \zeta_5 \times \zeta_6 \times \zeta_3 \times \zeta_3$$
$$- 2 \times \zeta_0 \times \zeta_1 \times \zeta_6 \times \zeta_7 \times \zeta_7 - 2 \times \zeta_1 \times \zeta_3 \times \zeta_7 \times \zeta_5 \times \zeta_5 - 2 \times \zeta_4 \times \zeta_6 \times \zeta_7 \times \zeta_2 \times \zeta_2$$
$$+ 2 \times \zeta_0 \times \zeta_2 \times \zeta_7 \times \zeta_6 \times \zeta_6 + 2 \times \zeta_2 \times \zeta_3 \times \zeta_4 \times \zeta_6 \times \zeta_6 + 3 \times \zeta_1 \times \zeta_2 \times \zeta_4 \times \zeta_7 \times \zeta_7$$
$$+ \zeta_3 \times \zeta_3 \times \zeta_5 \times \zeta_5 \times \zeta_5 + 3 \times \zeta_0 \times \zeta_0 \times \zeta_7 \times \zeta_7 \times \zeta_7 - 2 \times \zeta_0 \times \zeta_3 \times \zeta_6 \times \zeta_6 \times \zeta_6;$$

$$\gamma_y = \zeta_0 \times \zeta_2 \times \zeta_3 \times \zeta_5 \times \zeta_7 + \zeta_1 \times \zeta_2 \times \zeta_3 \times \zeta_5 \times \zeta_6 - \zeta_0 \times \zeta_1 \times \zeta_3 \times \zeta_6 \times \zeta_7$$
$$- 6 \times \zeta_1 \times \zeta_2 \times \zeta_3 \times \zeta_4 \times \zeta_7 - \zeta_1 \times \zeta_1 \times \zeta_1 \times \zeta_7 \times \zeta_7 - 3 \times \zeta_3 \times \zeta_3 \times \zeta_3 \times \zeta_4 \times \zeta_4$$
$$- \zeta_1 \times \zeta_3 \times \zeta_3 \times \zeta_5 \times \zeta_5 - \zeta_3 \times \zeta_1 \times \zeta_1 \times \zeta_6 \times \zeta_6 - 3 \times \zeta_3 \times \zeta_0 \times \zeta_0 \times \zeta_7 \times \zeta_7$$
$$+ \zeta_2 \times \zeta_6 \times \zeta_7 \times \zeta_1 \times \zeta_1 - \zeta_1 \times \zeta_5 \times \zeta_7 \times \zeta_2 \times \zeta_2 - 3 \times \zeta_0 \times \zeta_5 \times \zeta_6 \times \zeta_3 \times \zeta_3$$
$$- 2 \times \zeta_0 \times \zeta_6 \times \zeta_7 \times \zeta_2 \times \zeta_2 - 2 \times \zeta_3 \times \zeta_4 \times \zeta_6 \times \zeta_2 \times \zeta_2 + 2 \times \zeta_0 \times \zeta_2 \times \zeta_3 \times \zeta_6 \times \zeta_6$$
$$+ 2 \times \zeta_2 \times \zeta_4 \times \zeta_5 \times \zeta_3 \times \zeta_3 + 2 \times \zeta_3 \times \zeta_5 \times \zeta_7 \times \zeta_1 \times \zeta_1 + 3 \times \zeta_0 \times \zeta_1 \times \zeta_2 \times \zeta_7 \times \zeta_7$$
$$+ 4 \times \zeta_1 \times \zeta_4 \times \zeta_6 \times \zeta_3 \times \zeta_3 + 6 \times \zeta_0 \times \zeta_4 \times \zeta_7 \times \zeta_3 \times \zeta_3 + 2 \times \zeta_4 \times \zeta_7 \times \zeta_2 \times \zeta_2 \times \zeta_2;$$

$$\gamma_0 = \zeta_0 \times \zeta_1 \times \zeta_2 \times \zeta_5 \times \zeta_6 \times \zeta_7 + \zeta_0 \times \zeta_1 \times \zeta_3 \times \zeta_4 \times \zeta_6 \times \zeta_7 - \zeta_0 \times \zeta_2 \times \zeta_3 \times \zeta_4 \times \zeta_5 \times \zeta_7$$
$$- \zeta_1 \times \zeta_2 \times \zeta_3 \times \zeta_4 \times \zeta_5 \times \zeta_6 + \zeta_4 \times \zeta_1 \times \zeta_1 \times \zeta_1 \times \zeta_7 \times \zeta_7 - \zeta_7 \times \zeta_2 \times \zeta_2 \times \zeta_2 \times \zeta_4 \times \zeta_4$$
$$+ \zeta_1 \times \zeta_4 \times \zeta_3 \times \zeta_3 \times \zeta_5 \times \zeta_5 + \zeta_1 \times \zeta_6 \times \zeta_0 \times \zeta_0 \times \zeta_7 \times \zeta_7 + \zeta_3 \times \zeta_4 \times \zeta_1 \times \zeta_1 \times \zeta_6 \times \zeta_6$$
$$+ \zeta_3 \times \zeta_6 \times \zeta_2 \times \zeta_2 \times \zeta_4 \times \zeta_4 - \zeta_0 \times \zeta_5 \times \zeta_1 \times \zeta_1 \times \zeta_7 \times \zeta_7 - \zeta_0 \times \zeta_7 \times \zeta_2 \times \zeta_2 \times \zeta_5 \times \zeta_5$$
$$- \zeta_2 \times \zeta_5 \times \zeta_3 \times \zeta_3 \times \zeta_4 \times \zeta_4 - \zeta_2 \times \zeta_7 \times \zeta_0 \times \zeta_0 \times \zeta_6 \times \zeta_6 - 3 \times \zeta_0 \times \zeta_7 \times \zeta_3 \times \zeta_3 \times \zeta_4 \times \zeta_4$$
$$- 2 \times \zeta_1 \times \zeta_6 \times \zeta_3 \times \zeta_3 \times \zeta_4 \times \zeta_4 + 2 \times \zeta_2 \times \zeta_5 \times \zeta_0 \times \zeta_0 \times \zeta_7 \times \zeta_7$$
$$+ 3 \times \zeta_3 \times \zeta_4 \times \zeta_0 \times \zeta_0 \times \zeta_7 \times \zeta_7 + \zeta_0 \times \zeta_2 \times \zeta_3 \times \zeta_6 \times \zeta_5 \times \zeta_5 + \zeta_1 \times \zeta_4 \times \zeta_5 \times \zeta_7 \times \zeta_2 \times \zeta_2$$
$$- \zeta_0 \times \zeta_1 \times \zeta_3 \times \zeta_5 \times \zeta_6 \times \zeta_6 - \zeta_2 \times \zeta_4 \times \zeta_6 \times \zeta_7 \times \zeta_1 \times \zeta_1 - 3 \times \zeta_0 \times \zeta_1 \times \zeta_2 \times \zeta_4 \times \zeta_7 \times \zeta_7$$
$$- 3 \times \zeta_3 \times \zeta_5 \times \zeta_6 \times \zeta_7 \times \zeta_0 \times \zeta_0 - 2 \times \zeta_0 \times \zeta_2 \times \zeta_3 \times \zeta_4 \times \zeta_6 \times \zeta_6$$
$$- 2 \times \zeta_3 \times \zeta_4 \times \zeta_5 \times \zeta_7 \times \zeta_1 \times \zeta_1 + 2 \times \zeta_0 \times \zeta_1 \times \zeta_3 \times \zeta_7 \times \zeta_5 \times \zeta_5$$
$$+ 2 \times \zeta_0 \times \zeta_4 \times \zeta_6 \times \zeta_7 \times \zeta_2 \times \zeta_2 + 3 \times \zeta_0 \times \zeta_4 \times \zeta_5 \times \zeta_6 \times \zeta_3 \times \zeta_3$$
$$+ 3 \times \zeta_1 \times \zeta_2 \times \zeta_3 \times \zeta_7 \times \zeta_4 \times \zeta_4 + \zeta_3 \times \zeta_3 \times \zeta_3 \times \zeta_4 \times \zeta_4 \times \zeta_4 - \zeta_0 \times \zeta_0 \times \zeta_0 \times \zeta_7 \times \zeta_7 \times \zeta_7$$
$$+ \zeta_3 \times \zeta_0 \times \zeta_0 \times \zeta_6 \times \zeta_6 \times \zeta_6 - \zeta_0 \times \zeta_3 \times \zeta_3 \times \zeta_5 \times \zeta_5 \times \zeta_5;$$

Note that, while the above calculation may look complicated, essentially, through the above, each of the coefficients in Eq. 14 is computed over the coordinates of the four control points of the curve (i.e. $\{x_0, y_0, x_1, y_1, x_2, y_2, x_3, y_3\}$) just through a group of basic arithmetic operations. In other words, **the computation of the coefficients in Eq. 14 is just a task with $O(1)$ time complexity**.

## D   Additional Details about the Bézier-boundary Gradient Approximation Strategy

In our framework, we propose a Bézier-boundary gradient approximation strategy to keep the "differentiability" of the rendering process. In Sec. 4.2 in the main paper, we explain this strategy taking the approximation of $\frac{\partial g}{\partial c_{curve}^{2D}[0,0]}$ as an example. Here in this section, with $c_{curve} \in \mathbb{R}^{4M \times 2}$, we further describe how we approximate $\frac{\partial g}{\partial c_{curve}^{2D}[i,j]}$, where $i \in \{0, ..., 4M - 1\}$ and $j \in \{0, 1\}$. Specifically, in our framework, $\frac{\partial g}{\partial c_{curve}^{2D}[i,j]}$ is approximated in a similar manner as $\frac{\partial g}{\partial c_{curve}^{2D}[0,0]}$ except in the following two places.

**(1) Definition change of $g_{sc}^0(p)$.** To approximate $\frac{\partial g}{\partial c_{curve}^{2D}[i,j]}$, we first need to redefine $g_{sc}^0(p)$ as $g_{sc}(c_{curve}^{2D}[4m], c_{curve}^{2D}[4m+1], c_{curve}^{2D}[4m+2], c_{curve}^{2D}[4m+3]; p)$, where $m = i \% 4$. This is done for $g_{sc}^0(p)$ to accurately represent the Bézier curve w.r.t. $c_{curve}^{2D}[i,j]$.

**(2) Reformulation of Eq. 11.** Moreover, during approximating $\frac{\partial g}{\partial c_{curve}^{2D}[i,j]}$, for $S_\phi$ to be corrected derived, we also need to reformulate Eq. 11 according to the Bézier curve w.r.t. $c_{curve}^{2D}[i,j]$. Note that in the reformulated equation, $\phi$ should be used in place of $c_{curve}^{2D}[i,j]$.

With the above two changes made, we can seamlessly use the strategy introduced in Sec. 4.2 in the main paper to approximate $\frac{\partial g}{\partial c_{curve}^{2D}[i,j]}$.

# E    Experiments on 11 3D Scenes.

In Tab. 1 in the main paper, following [30, 49, 22], we evaluate our method on a total of 13 3D scenes from the Mip-NeRF360 [4], Tanks&Temples [31], and Deep Blending [23] datasets. Here, following [35], we also evaluate our method on another benchmark with 11 3D scenes from the above three datasets. As shown in Tab. 12, on this evaluation benchmark, our method can also achieve superior performance consistently, further demonstrating the efficacy of our proposed method.

Table 12: Performance comparison following the evaluation benchmark of [35].

| Method | Mip-NeRF360 Dataset | | | Tanks&Temples | | | Deep Blending | | |
|---|---|---|---|---|---|---|---|---|---|
| | SSIM↑ | PSNR↑ | LPIPS↓ | SSIM↑ | PSNR↑ | LPIPS↓ | SSIM↑ | PSNR↑ | LPIPS↓ |
| Scaffold-GS [35] | 0.848 | 28.84 | 0.220 | 0.853 | 23.96 | 0.177 | 0.906 | 30.21 | 0.254 |
| Ours | **0.885** | **29.58** | **0.158** | **0.866** | **24.96** | **0.120** | **0.907** | **30.42** | **0.199** |

# F    Licenses

We use the Tanks&Temples dataset [31] by following the license of here. We use the Mip-NeRF360 dataset [4] by following the Apache-2.0 license. Moreover, we use the Deep Blending dataset [23] by following the license of here. Besides, we use part of the code owned by Kerbl et al. [30] by following the license of here.

