# OpenReview forum: "DisC-GS: Discontinuity-aware Gaussian Splatting"
_NeurIPS.cc/2024/Conference — NeurIPS 2024 poster_

### Official Review · Reviewer_ncnd · 2024-06-26

**Soundness:** 4
**Presentation:** 2
**Contribution:** 2
**Rating:** 4
**Confidence:** 4

**Summary:**

This paper proposes a novel kernel function to model static scenes, addressing the difficulty of modeling high-frequency boundaries caused by the $r^2$ decay from center to edge of the Gaussian kernel. To tackle the smoothing decay of the Gaussian kernel, the authors first divide the Gaussian in screen space using M Bézier curves, introducing discontinuities in the kernel function. To address the gradient discontinuity issue caused by the segmented Gaussian, the authors propose the Bézier-boundary Gradient Approximation Strategy to approximate the gradients during backpropagation, ensuring stable optimization. Quantitative and qualitative experiments on real-world scenes demonstrate that DisC-GS can model high-frequency boundaries and achieve the best rendering quality.

**Strengths:**

I think the ideas proposed in this paper are sound and easy to follow. Using two Bézier curves to segment the Gaussian, making it better at modeling high-frequency boundaries, is very reasonable and much needed by the community. The comparison in Tab. 1 with numerous compelling baselines and the excellent rendering metrics are greatly appreciated. While achieving the best rendering quality, a balance with FPS was also maintained, as shown in Tab. 11, where the rendering speed did not significantly decrease. The ablations are also appreciated.

**Weaknesses:**

1. Too few qualitative comparisons and mismatched quantitative metrics. In Figs. 3 and 4, only two scenes from deep blending, two scenes from tanks&temples, and Mip360's room are shown. After carefully comparing the rendered images from corresponding views of Scaffold-GS, I found that although DisC-GS shows slightly better visual effects, the differences in rendering metrics, especially SSIM and LPIPS, should not be so significant. Hence, I have a few questions: Did DisC-GS use the same data as vanilla GS (without rerunning COLMAP)? What was the image resolution used for calculating the rendering metrics, and were the SSIM and LPIPS (VGG) calculations done in the same manner as for vanilla GS?
2. Lack of comparisons on synthetic scenes. In real-world scenes, there can be inaccuracies in camera poses, which may lead to improvements in metrics that are unrelated to the method itself. Therefore, the rendering quality on synthetic scenes would be more convincing. I am very much looking forward to seeing the rendering metrics and image quality of DisC-GS on the NeRF and NSVF datasets.
3. Lack of discussion on the number of Gaussians, training time and storage. I am very curious about the approximate number of Gaussians used in DisC-GS, as the number of Gaussians can greatly affect rendering quality.
4. I have some doubts about the optimization of DisC-GS. It seems that this optimization method could easily get stuck in local optima, and introducing polynomial computations in CUDA kernels might significantly reduce FPS. Did the authors use any special optimization strategies?

**Questions:**

I have some doubts about evaluation in this paper. However, due to the strong results and sound methodology, if the authors can address my concerns, I am very open to raising my score to `accept`.

**Limitations:**

Please refer to the weakness part.

---

> ### Author Rebuttal · Authors · 2024-08-07
>
> >*Q1: Qualitative comparisons and quantitative metrics.*
>
> **A1:** **(1) Qualitative comparisons.** In the PDF uploaded during rebuttal (at the bottom of the "Author Rebuttal by Authors" comment), besides in the 3D scenes in Figs 3 and 4 in paper, we also provide qualitative comparisons in more 3D scenes. Moreover, Scaffold-GS is also added in the comparisons. As shown, across different 3D scenes, our framework consistently achieves better visual effects than existing methods such as Scaffold-GS, further showing the efficacy of our framework. Due to the space limitation of the PDF, we'll also add qualitative comparisons in more 3D scenes to paper.
>
> **(2) Quantitative metrics.** Moreover, for the mentioned mismatch between the difference in rendering metrics and the difference in visual effects, this can be because, different testing views are of different complexities (i.e., containing different numbers and scales of discontinuities). During measuring the rendering metrics, all different testing views (including those very complex ones) are utilized and they all contribute to the measurement. Yet, in Figs 3 and 4 in paper which serve as the qualitative results of our method in general, we did not dedicatedly show all those very complex testing views that suffer from very severe boundary modeling issue.
>
> As shown in the PDF uploaded during rebuttal, while good visual effects can be consistently achieved by our framework across different viewpoints and different 3D scenes, especially over those testing views that are more complex (richer in boundaries), the difference in visual effects between our method and existing methods such as Scaffold-GS tends to be more significant. These complex testing views, during quantitative evaluation, also tends to more significantly contribute to the superior performance of our method in metrics such as SSIM and LPIPS. We will also add more qualitative comparisons over those complex testing views across different 3D scenes to paper.
>
> **(3) Questions w.r.t. evaluation details.** (3.1) Yes. We use the same data as vanilla GS without rerunning COLMAP. (3.2) Following the vanilla GS, during calculating the rendering metrics, for images with width exceeding 1600, we would rescale them for their width to be 1600. For rest images we would use their original resolution. (3.3) Yes, SSIM and LPIPS calculations are both done in the same manner as vanilla GS.
>
> >*Q2: Synthetic scenes.*
>
> **A2:** As suggested, we also evaluate our method on the synthetic NSVF and NeRF datasets, on an RTX 3090 GPU.
>
> For rendering metrics, below, we report the SSIM, PSNR, and LPIPS metrics. Moreover, we also show images rendered over these two datasets in the PDF uploaded during rebuttal.
> |Method|NSVF-PSNR↑|NSVF-SSIM↑|NSVF-LPIPS↓|NeRF-PSNR↑|NeRF-SSIM↑|NeRF-LPIPS↓|
> |-|-|-|-|-|-|-|
> |2D Gaussian Splatting|37.59|0.984|0.014|33.96|0.969|0.032|
> |Ours (on 2D Gaussian Splatting)|**38.37**|**0.988**|**0.010**|**34.18**|**0.973**|**0.025**|
> |3D Gaussian Splatting|37.07|0.987|0.015|33.32|0.970|0.031|
> |Ours (on 3D Gaussian Splatting)|**38.32**|**0.988**|**0.012**|**34.01**|**0.972**|**0.029**|
>
> As shown above, on synthetic scenes, our framework, when applied on both 2D and 3D Gaussian Splattings, can consistently achieve performance improvements. Moreover, as shown in the PDF, on synthetic scenes, our framework can also render images with high quality. This further shows our framework's efficacy.
>
> >*Q3: Discussion on the number of Gaussians, training time and storage.*
>
> **A3:** Below, we show the number of Gaussians, storage, training time, and inference time of our framework, on the Tanks&Temples dataset on an RTX 3090 GPU.
> |Method|PSNR↑|Number of Gaussians|Storage|Training time|Inference time (per image)|
> |-|-|-|-|-|-|
> |2D Gaussian Splatting|23.30|~1585K|376MB|0.27 hour|0.007s|
> |Ours (on 2D Gaussian Splatting)|24.96|~909K|299MB|0.32 hour|0.008s|
> |3D Gaussian Splatting|23.14|~1784K|423MB|0.27 hour|0.007s|
> |Ours (on 3D Gaussian Splatting)|24.67|~1094K|410MB|0.33 hour|0.008s|
>
> As shown, though our framework achieves obviously better performance, it uses less number of Gaussians, does not increase the memory storage, only brings a relatively small increase of training time, and can perform rendering efficiently during inference. We'll discuss the above in paper.
>
> >*Q4: Optimization of DisC-GS.*
>
> **A4:** **(1) Computations.** No, we do not use special optimization strategies. Instead, we would like to highlight that computations introduced by DisC-GS are not mathematically complex. Specifically, in the forward rendering process, as mentioned in lines 261-265 in paper, for each Bézier curve, the binary indicator function that DisC-GS builds for it is with $O(1)$ time complexity. Meanwhile, in the gradient backpropagation process, the cubic functions that DisC-GS introduces (Eq. 11 in paper) also have closed-form solutions, and can thus be very efficiently solved also in $O(1)$ time complexity. Thus, overall, as shown above in **A3**, our framework would only bring a small increase of training time. Meanwhile, during inference, our framework can also perform rendering efficiently.
>
> **(2) Local optima.** As mentioned above, the computations our framework introduce are not complex. It thus would not add significant complexity to the typical Gaussian Splatting technique. Specifically, across different 3D scenes among different datasets, our framework consistently outperforms both 2D and 3D Gaussian Splattings leveraging the typical Gaussian Splatting technique. This also implies that our framework would not easily get stuck in local optima. Meanwhile, in the PDF uploaded during rebuttal, we also provide loss curves over the 3D scene Train in the Tanks&Temples dataset. As shown, no matter when applied to 2D or 3D Gaussian Splattings, our framework consistently achieves a similar loss reduction trend. This further implies that our framework would not raise the risk of easily getting stuck in local optima.

---

> > ### Comment · Reviewer_ncnd · 2024-08-08
> > **Feedback from reviewer ncnd**
> >
> > Thanks for authors' response and the additional experimental results based on the review. However, I still do not understand why there is such a significant improvement in the quantitative metrics (tanks from 0.178 to 0.120, db from 0.240 to 0.199) based on the authors' statement that they used the same resolution and rendering metrics as vanilla 3D-GS. I have the following questions:
> > - Fig. 3(b)(c) shows a significant reduction in floaters, but I have no idea about the relationship between this and DisC-GS itself. The essence of DisC-GS is to address the need for extensive boundary modeling in 3D-GS, which is unrelated to floater removal. The paper should clearly state what causes the removal of floaters instead of obscuring the components that actually contribute to this effect.
> > - Fig. 3(a) presents slightly better visual results compared to Scaffold-GS, but it does not sufficiently support the improvement of LPIPS (vgg) from 0.177 to 0.120 in the tanks scene.
> > - The paper does not show changes in the distribution of GS. DisC-GS theoretically reduces the number of GS required for modeling high-frequency boundaries, so the reduction in quantity and memory usage makes sense. However, the lack of visualization of the point cloud distribution is quite confusing.
> > - While DisC-GS theoretically addresses the ability of 3D-GS to model high-frequency boundaries, is it really always better than 3D-GS? In many scenes (e.g., diffuse scenes), we actually need the smooth kernel provided by 3D-GS.
> >
> > Overall, this paper still lacks sufficient evaluation to demonstrate the advantages of DisC-GS relative to vanilla 3D-GS (excluding improvements beyond changes in the kernel function). Therefore, I still maintain a negative evaluation of this work.

---

> > > ### Author Response · Authors · 2024-08-13
> > > **Response to reviewer ncnd [1/2]**
> > >
> > > Thank you for your time and effort. Below, we would like to answer your follow-up questions.
> > >
> > > >*Q5: Fig. 3(b)(c) shows a significant reduction in floaters, but I have no idea about the relationship between this and DisC-GS itself. The essence of DisC-GS is to address the need for extensive boundary modeling in 3D-GS, which is unrelated to floater removal. The paper should clearly state what causes the removal of floaters instead of obscuring the components that actually contribute to this effect.*
> > >
> > > **A5:** We carefully investigate Fig. 3(b)(c) and their corresponding 3D scene representations. We observe that, in the rendered images of existing methods in Fig. 3(b)(c), a large number of continuous Gaussian kernels are messily stacked in areas with rich boundaries and discontinuities. This stacking leads to floaters and blurriness, particularly in the red-boxed areas of these figures. We emphasize that this issue arises from a fundamental limitation of Gaussian Splatting, which our DisC-GS framework aims to address. Specifically, due to the continuous nature of Gaussian distributions, Gaussian Splatting struggles to accurately render discontinuities and boundaries in images (see Lines 3-5 in the paper). As a result, existing Gaussian Splatting methods often produce low-quality renderings in boundary-rich areas, with noticeable blurriness and floaters. The above points out that, **our floater reduction in Fig. 3(b)(c) is closely related to the essence of our DisC-GS framework**. We will also discuss this in paper.
> > >
> > >
> > > >*Q6: Fig. 3(a) presents slightly better visual results compared to Scaffold-GS, but it does not sufficiently support the improvement of LPIPS (vgg) from 0.177 to 0.120 in the tanks scene.*
> > >
> > > **A6:** (1) In **Fig. 3(a)**, our DisC-GS framework only slightly improves LPIPS over Scaffold-GS, i.e., from 0.188 to 0.179 by 0.009. This is consistent with the slight visual result improvement of DisC-GS over Scaffold-GS in Fig. 3(a).
> > >
> > > (2) **Yet, we highlight that, the improvement of LPIPS from 0.177 to 0.120 in the tanks dataset is still sufficiently supported by its testing images**. This is because, in many other testing images in the tanks dataset, DisC-GS achieves much more significant LPIPS improvements over Scaffold-GS. For example, in the 10th testing image in the *Train* scene in this dataset, DisC-GS improves LPIPS over Scaffold-GS from 0.211 to 0.110 by 0.101; in the 13th testing image, DisC-GS improves LPIPS over Scaffold-GS from 0.203 to 0.101 by 0.102. Note that, the *Train* scene in the above sentence refers to the Train (railway) scene, but not the scene used for training. Meanwhile, we also observe that, in these testing images that DisC-GS achieves much more significant LPIPS improvements, DisC-GS can also achieve much more significant visual result improvements. So overall the improvement of LPIPS on the whole dataset is obvious. Thanks for this comment. Besides Fig. 3(a), we will also add these testing images in paper.

---

> > > > ### Author Response · Authors · 2024-08-13
> > > > **Response to reviewer ncnd [2/2]**
> > > >
> > > > >*Q7: The paper does not show changes in the distribution of GS. DisC-GS theoretically reduces the number of GS required for modeling high-frequency boundaries, so the reduction in quantity and memory usage makes sense. However, the lack of visualization of the point cloud distribution is quite confusing.*
> > > >
> > > > **A7:** Below, between our DisC-GS framework and 3D-GS, we compare the number (distribution) of Gaussian kernels, per unit volume ($1\times1\times1$) in the xyz 3D coordinate system of the 3D scene. Specifically, we perform comparisons in the edge regions of the 3D scene that are detected utilizing the Canny edge detection algorithm [9], as well as in the non-edge regions of the 3D scene, on the Tanks&Temples dataset.
> > > >
> > > > |Method|Number of Gaussian kernels per unit volume in edge regions of the 3D scene|Number of Gaussian kernels per unit volume in non-edge regions of the 3D scene|
> > > > |---|---|---|
> > > > | Original 3D-GS | 0.98 | 0.39 |
> > > > | Ours (on 3D-GS) | 0.31 | 0.30 |
> > > >
> > > > As shown, compared to the original 3D-GS, our DisC-GS framework can represent the 3D scene with much fewer Gaussian kernels per unit volume, especially in the edge regions of the scene that contain a large number of high-frequency boundaries. This further experimentally shows that, DisC-GS can reduce the number of Gaussian kernels required for modeling high-frequency boundaries. Due to the format constraint here, we will also provide the visualization of the point cloud distribution in paper to further show changes in the distribution of Gaussian kernels.
> > > >
> > > > >*Q8: While DisC-GS theoretically addresses the ability of 3D-GS to model high-frequency boundaries, is it really always better than 3D-GS? In many scenes (e.g., diffuse scenes), we actually need the smooth kernel provided by 3D-GS.*
> > > >
> > > > **A8:** (1) **Yes, DisC-GS is always better than 3D-GS**. To investigate this, we measure the per-image metrics for all testing images in all 13 evaluated 3D scenes. We find that, in all testing images in different 3D scenes, DisC-GS can always perform better than 3D-GS. In specific, in all testing images, the [minimal per-image improvement, maximal per-image improvement] range of DisC-GS over 3D-GS is [+0.01, +9.64] for the PSNR metric, [+0.001, +0.059] for the SSIM metric, and [+0.002, +0.121] for the LPIPS metric.
> > > >
> > > > (2) We agree that, we still need smooth kernels in many scenes. Yet, we highlight that, **similar to 3D-GS, if needed, DisC-GS also can contain "smooth" kernels**. This is because, in DisC-GS, for each Gaussian kernel, the places where it is scissored over are learnable (see Lines 218-221 in paper). Thus, in DisC-GS, when a certain Gaussian kernel still needs to be a smooth kernel, DisC-GS is also able to learn to scissor this kernel only in places very far from its center. This leads the scissoring to have negligible effects on this kernel, and thus still keeps this kernel a "smooth" kernel. Yet, note that, different from 3D-GS, in DisC-GS, besides "smooth" kernels, DisC-GS also contains non-smooth kernels that can represent boundaries and discontinuities. This leads to the consistent performance improvements of DisC-GS over 3D-GS (see (1)).

---

### Official Review · Reviewer_Kjj1 · 2024-07-11

**Soundness:** 3
**Presentation:** 3
**Contribution:** 3
**Rating:** 7
**Confidence:** 4

**Summary:**

This paper introduces DisC-GS, a method that utilizes Bezier curves for discontinuity-aware image rendering on 3DGS. By employing Bezier curves, this approach significantly enhances the rendering results of scene boundaries. A set of experiments and ablation studies substantiate the effectiveness of this proposed method.

**Strengths:**

1. The author thoroughly discusses the discontinuous characteristics of GS, which lead to inaccurate drawing of boundary curves in images. Innovatively, they propose using Bezier curves to draw continuous boundaries.
2. The experimental results are highly convincing, demonstrating the superiority of the proposed method over state-of-the-art algorithms, and are supported by abundant ablation experiments.
3. The authors’ clear writing, well-structured discussion of the problem, detailed methods, and comprehensive presentation of experimental results make the paper easy to understand.

**Weaknesses:**

1. Introducing Bezier curves to refine Gaussian image rendering is indeed very clever. However, this paper only uses 2D parameters to control the position of the control points. Could this cause rendering problems in areas significantly affected by large viewpoint changes?
2. When additional properties such as control points and curve rendering are introduced, what impact does this have on training efficiency? While there is a time comparison for real-time rendering in the appendix, it would be interesting to know whether adding these parameters has any additional impact on training time and convergence (number of iterations).
3. Given that the authors changed the properties of each Gaussian, it would be interesting to see how these modifications affect the distribution of Gaussians in edge and surface regions. One aspect that could be explored is whether the number of Gaussians in the scene will differ significantly from the original 3DGS.

**Questions:**

See the weakness part.

**Limitations:**

This paper lacks a discussion of limitations. It is recommended to provide a detailed description of the challenges associated with large scenes, the handling of huge parameters, posed images, and observation views that require improvement in reconstruction to ensure completeness.

---

> ### Author Rebuttal · Authors · 2024-08-07
>
> >*Q1: 2D parameters to control the position of the control points.*
>
> **A1:** As mentioned in lines 121-125 in the paper, in existing works, both 2D and 3D Gaussian Splattings have been utilized to represent the 3D scene. In this work, we proposed a framework that can be applied to both 2D and 3D Gaussian Splattings. Specifically, as mentioned in lines 218-220 and lines 388-392 in the paper, **when applying our framework on 2D Gaussian Splatting, we use 2D parameters to control the positions of the control points, while when applying our framework on 3D Gaussian Splatting, we instead use 3D parameters**. As shown in Tab. 2 in the paper, no matter when applied to 2D or 3D Gaussian Splattings, our framework can consistently achieve performance improvements. Meanwhile, in both cases, we do not observe rendering problems in areas significantly affected by large viewpoint changes. We will further clarify the above in the paper to avoid confusion.
>
> >*Q2: When additional properties such as control points and curve rendering are introduced, what impact does this have on training efficiency? [...] impact on training time and convergence(number of iterations).*
>
> **A2:** **(1) Training time.** Below, similarly to the rendering time shown in the Appendix, we also show the training time of our framework, on the Tanks&Temples dataset on an RTX 3090 GPU.
>
> | Method | PSNR↑ | Training time |
> |-|-|-|
> | Mip-NeRF360 | 22.22 | 48 hours |
> | 2D Gaussian Splatting | 23.30 | 0.27 hour |
> | 3D Gaussian Splatting | 23.14 | 0.27 hour |
> | Ours (on 2D Gaussian Splatting) | 24.96 | 0.32 hour |
> | Ours (on 3D Gaussian Splatting) | 24.67 | 0.33 hour |
>
> As shown, though our framework achieves obviously better performance, no matter when applied to 2D or 3D Gaussian Splattings, our framework only brings a relatively small increase of training time.
>
> **(2) Convergence (Number of iterations).** Meanwhile, in the PDF uploaded during rebuttal (at the bottom of the "Author Rebuttal by Authors" comment), we also provide loss curves over the 3D scene Train in the Tanks&Temples dataset. As shown, our framework consistently achieves a similar convergence rate compared to the baseline without parameters such as the control points added. Moreover, for a fair comparison, in the experiments in the paper, we train our framework for the same number of iterations (i.e., 30k iterations) as the existing Gaussian Splatting works.
>
> We will add the above discussion on training time and convergence to the paper.
>
> >*Q3: Affects on the distribution of Gaussians in edge and surface regions. [...] whether the number of Gaussians in the scene will differ significantly from the original 3DGS*
>
> **A3:** Below, we aim to compare the total number of Gaussians in the whole 3D scene, as well as the number (distribution) of Gaussians in the edge and plane regions of the 3D scene, of our framework with the original 3DGS. Yet, both the edge regions and the plane regions of the 3D scene are not annotated in the dataset. Thus, to enable the latter of the above comparisons, below, we first describe how we estimate the edge regions and the plane regions of each 3D scene.
>
> Specifically, given a 3D scene, to estimate its edge regions, we here conduct the following 4 steps: (E.1) for each testing image of the scene, we first pass the image over the Canny algorithm [9] for detecting the pixels that are on the edge regions of the image. (E.2) After that, each testing image is further passed over the Depth Anything model [a] to acquire the depth values of its edge-region pixels detected in (E.1). (E.3) Then for each edge-region pixel detected in (E.1), leveraging its 2D coordinate, its depth value, and the viewpoint of its corresponding testing image, we can map this pixel back to the 3D space of the 3D scene. (E.4) Finally, we can define the union of the 3D positions of all the edge-region pixels as the estimated edge regions of the given 3D scene.
>
> Meanwhile, to estimate the plane regions of a 3D scene, we here conduct the following 3 steps: (P.1) for each testing image of the scene, we first pass it over the Depth Anything model [a] to acquire its depth map. (P.2) Given the depth map of a testing image, following [b], using Hough Transform, we can detect the pixels that are on the plane regions of the image. (P.3) Then similar to (E.3-E.4) above, we can transform all the plane-region pixels to the space of the 3D scene, and we can then finally define the union of the 3D positions of all the plane-region pixels as the estimated plane regions of the 3D scene.
>
> With the edges and the plane regions of the 3D scenes estimated in the above way, we can then estimate the number of Gaussians that cover (overlap with) these regions and thus perform comparisons over the number of Gaussians in the edge and plane regions of the 3D scene. Below, we compare the total number of Gaussians in the whole 3D scene, as well as the number of Gaussians in the edge and plane regions of the 3D scene, between our framework and the original 3DGS, on the Tanks&Temples dataset.
>
> |Method|Number of Gaussians in edge regions|Number of Gaussians in plane regions|Total number of Gaussians in the whole scene|
> |-|-|-|-|
> | Original 3DGS | ~605K | ~647K | ~1784K |
> | Ours (on 3DGS) | ~196K | ~433K | ~1094K |
>
> As shown, compared to the original 3DGS, our framework can represent the 3D scene with much fewer Gaussians, especially over its edge and plane regions. This further demonstrates our framework's ability in performing discontinuity-aware rendering and handling boundary issues. We will also discuss the above more in paper.
>
> [a] Depth Anything: Unleashing the Power of Large-Scale Unlabeled Data.
>
> [b] Hough Transform for real-time plane detection in depth images.
>
> >*Q4: Description of the challenges.*
>
> **A4:** Thanks for your suggestion. We agree that the Gaussian Splatting technique still faces challenges. As suggested, we will provide a detailed description of the mentioned challenges in the paper.

---

> > ### Comment · Reviewer_Kjj1 · 2024-08-09
> >
> > I have carefully reviewed the authors' responses and the supplementary material provided in the rebuttal.
> >
> > **Concerning A1**: The authors have not fully addressed my concern regarding the utilization of Bezier curves to shape the Gaussian distributions in 3D space. While Figure 2 in the manuscript illustrates the use of two Bezier curves to maintain the sharpness of the Gaussian on a plane, the methodology for employing Bezier curves to confine a 3D Gaussian within three-dimensional space remains inadequately described, both in the original manuscript and in the rebuttal.
> >
> > **Regarding A2 and A3**: The authors' method does enhance image rendering quality and reduces the number of Gaussians required, without a significant increase in training time.
> > My previous Q3 was about whether your method shows a significant performance difference between edge and non-edge areas.
> >
> > |  | Edge | Non-edge | Total |
> > | --- | --- | --- | --- |
> > | Original 3DGS | ~605K | ~1179K | ~1784K |
> > | Proposed Method on 3DGS | ~196K | ~898K | ~1094K |
> >
> > The significant reduction in the number of Gaussians used in edge regions (approximately 1/3 of the original count) suggests a potential enhancement in edge regions' representation. I recommend that the authors conduct additional experiments to compare the number of Gaussians and the resulting image render quality, specifically in edge regions. This could further substantiate the benefits of your approach and strengthen its appeal to others.

---

> > > ### Author Response · Authors · 2024-08-13
> > >
> > > Thank you for your time and effort in reviewing our work. Below, we further answer your following questions.
> > >
> > > >*Concerning A1: [...] the methodology for employing Bézier curves to confine a 3D Gaussian within three-dimensional space remains inadequately described.*
> > >
> > > **Answer:** To clarify, in our framework, we do not scissor (confine) 3D Gaussians within 3D space. Instead, even for 3D Gaussian Splatting, as mentioned in Lines 240-242 in paper, our framework always scissors (confines) 2D Gaussians that have been projected on the image plane. This is because, in our framework, the final goal is to modify the $\alpha$-blending function and thus enable the function to perform discontinuity-aware image rendering (see Lines 85-87 in paper). Note that, even for 3D Gaussian Splatting, its $\alpha$-blending function is also calculated based on the 2D Gaussians that have been projected on the image plane (see Eq. 2 in paper). Thus, in our framework, to effectively and conveniently modify the $\alpha$-blending function, we directly scissor 2D Gaussians on the image plane. We will make the above clearer in paper to avoid confusion.
> > >
> > > >*Regarding A2 and A3: My previous Q3 was about whether your method shows a significant performance difference between edge and non-edge areas. [...] The significant reduction in the number of Gaussians used in edge regions (approximately 1/3 of the original count) suggests a potential enhancement in edge regions' representation. I recommend that the authors conduct additional experiments to compare the number of Gaussians and the resulting image render quality, specifically in edge regions. This could further substantiate the benefits of your approach and strengthen its appeal to others.*
> > >
> > > **Answer:** (1) Yes, our method shows a significant performance difference in the edge and non-edge areas of the testing images. As shown below on the Tanks&Temples dataset, in the edge areas of its testing images, our method can achieve a much more significant performance improvement compared to 3DGS. This further shows the effectiveness of our method over edge areas of the image. Note that here, to perform evaluation effectively over image sub-areas, following [40], we use the MaskedSSIM metric (the larger the better).
> > >
> > > |Method|Number of Gaussians in edge areas of the scene|MaskedSSIM in edge areas of testing images|Number of Gaussians in non-edge areas of the scene|MaskedSSIM in non-edge areas of testing images|
> > > |---|---|---|---|---|
> > > | Original 3DGS | ~605K | 0.802 | ~1179K | 0.922 |
> > > | Ours (on 3DGS) | ~196K | 0.865 | ~898K | 0.928 |
> > > |Performance gain||+0.063||+0.006|
> > >
> > > (2) As suggested, below, we also compare the number of Gaussians and the resulting image render quality, in edge areas on the Tanks&Temples dataset. As shown, in our framework, with the increase of the number of Bézier curves we equip for each Gaussian, the number of Gaussians in edge areas of the scene consistently decreases. At the same time, in the edge areas of the testing images, our framework's image render quality consistently improves. In paper, taking our framework's efficiency into consideration, we equip 3 Bézier curves for each Gaussian (see Line 371 in paper).
> > >
> > > |Method|Number of Gaussians in edge areas of the scene|MaskedSSIM in edge areas of testing images|Performance gain over 3DGS|
> > > |---|---|---|---|
> > > | 0 Bézier curve for each Gaussian (Original 3DGS)  | ~605K | 0.802 ||
> > > | 1 Bézier curve for each Gaussian | ~324K | 0.838 |+0.036|
> > > | 2 Bézier curves for each Gaussian | ~230K | 0.852 |+0.050|
> > > | 3 Bézier curves for each Gaussian | ~196K | 0.865 |+0.063|
> > > | 4 Bézier curves for each Gaussian | ~191K | 0.866 |+0.064|
> > >
> > > We will also add the above experiments to paper.

---

> ### Comment · Reviewer_Kjj1 · 2024-08-13
>
> Thank you to the author for providing additional explanation.
>
> So, if we assume the presence of 4M control points, are these points situated within a 3D space and projected onto a 2D image plane?
> If this is the case, it appears that the impact on 2DGS might be minimal. However, for 3DGS, it seems that the method might inadvertently reduce 3DGS to a quasi-2D state.
>
> Additionally, the MaskedSSIM comparison offers a clearer measure of improvement in the edge areas.

---

> > ### Author Response · Authors · 2024-08-14
> >
> > Thanks for your timely response.
> >
> > **(1)** Yes, in our framework, the 4M control points are first situated in a 3D space and then projected onto a 2D image plane.
> >
> > **(2)** We're glad you think 2DGS appears to be compatible to this design. Below, we highlight that, with this design, when applying our framework on 3DGS, we can also achieve a good novel-view image rendering quality. This is because, as mentioned in existing works like [29], for Gaussian Splatting to perform good and detailed novel-view synthesis, in its (experimental) setting, Gaussian Splatting naturally expects the pose (angle) differences between the novel testing views and their surrounding training views to be not large. In this case with non-large view differences, even though we handle 3DGS in a quasi-2D manner (i.e., contouring scene boundaries on the image planes of the training views instead of directly in the 3D space), over the novel testing views, our framework can still acquire a good representation of scene boundaries.
> >
> > Meanwhile, we highlight that, rather than handling 3DGS in a quasi-2D manner, directly scissoring Gaussian kernels into non-Gaussian kernels in the 3D space and then projecting (splatting) these kernels onto the image plane can be a design that is not compatible to Gaussian Splatting. This is because, as mentioned in [52], while the splatting (projection) of Gaussian kernels approximately holds the closed-form solution (see Eq. 1 in paper) and can thus be easily performed, the splatting (projection) of non-Gaussian kernels can be a very difficult problem.
> >
> > In summary, in our framework, from an innovative perspective, we propose to perform post-scissoring after Gaussian kernels have been projected onto the image plane. This can bypass the difficulty of splatting (projecting) non-Gaussian kernels, while at the same time can also enable scene boundaries to be effectively represented and rendered. As shown in Table 2 in paper, this also leads the applications of our framework on both 2DGS and 3DGS to consistently achieve good performance.
> >
> > We will also discuss the above in paper.

---

### Official Review · Reviewer_F99C · 2024-07-11

**Soundness:** 4
**Presentation:** 3
**Contribution:** 4
**Rating:** 5
**Confidence:** 4

**Summary:**

This paper proposes DisC-GS, a technique that enhances the boundary rendering quality for Gaussian splatting. DisC-GS takes into account the discontinuity of shapes and uses Bézier curves to model the boundaries. To enable differentiable rendering, the authors propose a novel discontinuity-aware rendering pipeline paired with a gradient approximation strategy. Experiments show that DisC-GS surpasses existing methods in rendering quality.

**Strengths:**

This paper addresses a very important problem regarding the representability of 3DGS. It introduces a reasonable pipeline to tackle this issue. The idea of using Bézier curves to define the shape is novel and sound.

DisC-GS encodes Bézier curves as an additional attribute of Gaussians, and the rendering and training schemes are compatible with the original 3DGS. This compatibility means that this method can be easily adopted by most 3DGS-based methods.

Furthermore, this paper proposes an effective gradient approximation strategy, making the training of this representation feasible.

**Weaknesses:**

DisC-GS will increase the storage of the 3D scene representaion. The authors did not evaluate this in experiments.

The authors do not provide additional qualitative results, such as a video with continuous camera movements. It would be interesting to see if DisC-GS can preserve multi-view consistency. DisC-GS produces hard boundaries for Gaussians, which may cause artifacts (e.g., flickering) in such videos.

The methods compared in the paper are not very comprehensive; for example, there are no results compared with mip-splatting. The aliasing artifacts demonstrated in Figure 3 may not be entirely due to boundary issues, and more comparisons are needed to prove this.

**Questions:**

If I understand correctly, the Bézier curves for each Gaussian are only defined in a 2D subspace. Is there a specific reason for this design? Can they be directly defined in a three-dimensional space? Is this design similar to 2D Gaussian splatting?

**Limitations:**

The authors do not provide any limitations.  There may be a limitation about the storage size.

---

> ### Author Rebuttal · Authors · 2024-08-07
>
> >*Q1: Evaluation of the storage.*
>
> **A1:** Below, we show the storage of our framework on the Tanks&Temples dataset on an RTX 3090 GPU. As shown, either applied on 2D or 3D Gaussian Splattings, our framework **does not increase the storage size**.
>
> This can be because, admittedly, for each Gaussian representing the 3D scene, our framework equips it with a fifth attribute to represent the Bézier curves and thus makes it have slightly more parameters. Yet, on the other hand, with this additional attribute incorporated, our framework also enables each Gaussian representing the 3D scene to be aware of the discontinuity of shapes and thus enhance its representation ability. As shown below, this allows our framework to ultimately represent the 3D scene with much fewer Gaussians, thus not incurring the increase of memory storage. We'll discuss this more in paper.
> |Method|Storage|Total number of Gaussians|
> |-|-|-|
> |2D Gaussian Splatting|376MB|~1585K|
> |Ours (on 2D Gaussian Splatting)|299MB|~909K|
> |3D Gaussian Splatting|423MB|~1784K|
> |Ours (on 3D Gaussian Splatting)|410MB|~1094K|
>
> >*Q2: Additional qualitative results.*
>
> **A2:** (1) As suggested, we have rendered additional qualitative results (i.e., videos over different 3D scenes each with continuous camera movements). Among these videos, we observe that DisC-GS can consistently preserve multi-view consistency. Meanwhile, in these videos, artifacts such as flickering are also not observed. This can be because, while DisC-GS produces hard boundaries, the boundaries produced over different viewpoints are all based on the projection of the same sets of control points stored in the Gaussians representing the 3D scene. Thus, (multi-view) consistency of boundaries across different views can be kept. Meanwhile, with boundaries rendered in a multi-view consistent manner, artifacts such as flickering are also not observed. (2) During rebuttal, due to its format and space constraint, we provide a sample of such video qualitative results in GIF format in the PDF at the bottom of the "Author Rebuttal by Authors" comment. To view those results in animation mode, please use a computer and Adobe Acrobat Reader Version 2024.002 (downloadable from Adobe Reader's official website). We'll also include more results in paper.
>
> >*Q3: Comparison with mip-splatting.*
>
> **A3:** Below, we also compare our method with mip-splatting. As shown, on all three metrics and across various datasets, our framework consistently outperforms mip-splatting. This further shows the efficacy of our framework. We'll include mip-splatting in Tab. 1 in paper.
>
> |Method|Tanks&Temples-PSNR↑|Tanks&Temples-SSIM↑|Tanks&Temples-LPIPS↓|Mip-NeRF360-PSNR↑|Mip-NeRF360-SSIM↑|Mip-NeRF360-LPIPS↓|Deep Blending-PSNR↑|Deep Blending-SSIM↑|Deep Blending-LPIPS↓|
> |-|-|-|-|-|-|-|-|-|-|
> | Mip-splatting | 23.78 | 0.851 | 0.178 | 27.79 | 0.827 | 0.203 | 29.69 | 0.904 |  0.248 |
> | Ours  | **24.96** | **0.866** |  **0.120** | **28.01** | **0.833** |  **0.189** | **30.42** | **0.907** |  **0.199** |
>
> >*Q4: More comparisons w.r.t. boundary issues.*
>
> **A4:** In Figs 3 and 4 in paper, we demonstrate that, compared to both 2D and 3D Gaussian Splattings, our method can mitigate aliasing artifacts and achieve better rendering quality, especially in image regions containing numerous boundaries. To more clearly show that such advantages of our framework are due to its ability in handling boundary issues, we also perform quantitative comparisons.
>
> In Tab. 4 in paper (re-shown below), we quantitatively test our framework, particularly over the boundary-rich areas of the testing images. Specifically, for each testing image, to acquire its boundary-rich areas, we first use the Canny algorithm [9] to detect its boundaries. We then define its areas that involve or surround its Canny-detected boundaries as its boundary-rich areas, and define its rest areas as its boundary-sparse areas. To enable evaluation over image sub-areas, following [40], we use the MaskedSSIM metric (the larger the better). As shown, especially in the boundary-rich areas of the testing images, our framework can achieve a significant performance improvement. This shows our framework's ability to handle boundary issues from one perspective.
> |Method|Boundary-rich areas|Boundary-sparse areas|
> |-|-|-|
> |Baseline(2D Gaussian Splatting)|0.819|0.922|
> |Ours|0.855|0.934|
>
> Meanwhile, in Tab. 6 in paper (re-shown below), to evaluate if our framework can render sharp boundaries accurately instead of rendering them with blurriness, we also test our framework from the image sharpness perspective. Following [15], we measure image sharpness using the energy gradient function. As shown, our framework can increase the sharpness of its rendered images. This also from another perspective implies our framework's ability in accurately rendering the sharp boundaries in the image.
> |Method|Image sharpness|
> |-|-|
> |Baseline(2D Gaussian Splatting)|51.50%|
> |Ours|57.72%|
>
> We will further clarify the above in paper.
>
> >*Q5: Bézier curves in a 2D subspace.*
>
> **A5:** (1) In DisC-GS, we can define the Bézier curves in a 2D subspace, as well as directly in a three-dimensional space (3D space). (2) In the method section, we define the Bézier curves in a 2D subspace since we there take 2D Gaussian Splatting as an example and explain how we apply our framework on 2D Gaussian Splatting (as mentioned in lines 168-169 in paper). (3) We can also similarly define the Bézier curves in a 3D space for 3D Gaussian Splatting. As mentioned in lines 388-392 in paper, to achieve this, we just need to modify a single place of our framework (i.e., defining the control points of the Bézier curves for each Gaussian in the 3D space coordinate system instead). (4) As shown in Tab. 2 in paper, no matter when defining the Bézier curves in the 2D subspace or the 3D space, our framework can consistently achieve performance improvements. We will further clarify this in paper.

---

> > ### Comment · Reviewer_F99C · 2024-08-09
> >
> > Thanks to the authors for their response and the additional experimental results. Through their response, I realized that there are both 2D and 3D versions of Bézier curves. Even with this information, I believe that the description of the 3D version of Bézier curves in the paper is insufficient. It might be necessary to reorganize the methodology structure and provide some 3D illustrations to better explain it. Additionally, visualizing the learned Bézier curves could potentially offer a clearer explanation of why storage hasn't changed much, rather than vaguely attributing it to 'being aware of the discontinuity' and 'enhancing its representation ability.'

---

> > > ### Author Response · Authors · 2024-08-13
> > >
> > > Thank you for your time and effort and also thanks for your suggestions.
> > >
> > > **(1)** In our original paper, to ease readers' understanding, we first described the 2D version of Bézier curves as an example. After that, we described how the 3D version of Bézier curves can be similarly used in our framework (in Lines 388-392 in paper). Thank you for the suggestion, **to more sufficiently describe the 3D version of Bézier curves in our paper**, we will reorganize the methodology structure by adding a subsection at the end of the method section, as follows:
> > >
> > > 1. Specifically, in this subsection at the end of the method section, we will state that:
> > >
> > > Above, we focus on describing how we use 2D Bézier curves in our DisC-GS framework and correspondingly apply DisC-GS on 2D Gaussian Splatting. Here in this subsection, we further describe how we use 3D Bézier curves in our DisC-GS framework. Specifically, the transition from 2D to 3D Bézier curves in DisC-GS requires only two minimal modifications. (1) Firstly, for each Gaussian representing the 3D scene, the control points of its Bézier curves are stored directly in the 3D spatial coordinate system rather than in a 2D subspace.  Note that, this modification can be very simply made. Specifically, for each Gaussian in the 3D space in our DisC-GS framework, we only need to use $c_{curve}^{3D} \in \mathbb{R}^{4M\times3}$ instead of $c_{curve} \in \mathbb{R}^{4M\times2}$ to represent the control point coordinates of its 3D Bézier curves. In other words, for each 3D Gaussian, we only need to introduce it with $c_{curve}^{3D}$ instead of $c_{curve}$ as its new attribute. (2) Moreover, since we already directly introduce $c_{curve}^{3D}$ as the new attribute for each 3D Gaussian in our framework, during rendering, we omit Eq. 4 above in Sec. 4.1, which originally is used to acquire $c_{curve}^{3D}$ from $c_{curve}$. Overall, the above two modifications are sufficient to incorporate DisC-GS with 3D instead of 2D Bézier curves.
> > >
> > > 2. Moreover, in this subsection, we will also draw a figure to better explain the usage of 3D Bézier curves in our framework.
> > >
> > > Specifically, we will draw this new figure in a similar way as the current Figure 2 in paper. That is, in this new figure, we will include three sub-figures. Among these three sub-figures, in sub-figure (a), we will draw a 3D coordinate system and demonstrate the control points of the 3D Bézier curves in 3D space. In sub-figure (b), we will draw an image plane, on which we will draw the control points of the Bezier curves that have been projected on that image plane. Finally, in sub-figure (c), similar to the sub-figure (c) in the current Figure 2, we will demonstrate how Bézier curves are used in our framework to scissor the Gaussian distribution.
> > >
> > > We hope that this new subsection would facilitate the 3D version of Bézier curves to be more sufficiently described in our paper.
> > >
> > > **(2)** Moreover, we agree that the visualization of the learned Bézier curves can help to better explain why our framework does not increase the storage size. We will follow your suggestion to add this to our paper.

---

### Official Review · Reviewer_TofM · 2024-07-13

**Soundness:** 4
**Presentation:** 3
**Contribution:** 3
**Rating:** 5
**Confidence:** 4

**Summary:**

The authors proposed an innovative framework DisC-GS, which enables Gaussian Splatting to represent and render boundaries and discontinuities in an image, they also introduce several designs to make the pipeline is discontinuity-aware and differentiability, and their method achieves superior performance on the evaluated benchmarks

**Strengths:**

1. The authors propose a robust and sound pipeline that effectively emphasizes discontinuities and boundary situations while maintaining differentiability.

2. The proposed method demonstrates superior performance on the evaluated benchmarks

3. The paper presents a well-reasoned argumentation and reasoning process

**Weaknesses:**

The paper defines numerous labels, but they are presented in a noisy and unclear manner. It would be better to include a list of labels and their explanations. Additionally, I have to refer back to the main manuscript frequently as the lack of label explanations under each equation.

The paper does not include an analysis of the computational complexity of the proposed method.

There is no explanation or discussion on how the curves affect the densification procedure. For example, will the new cloned/splitted points are assigned the same curve attribute?

**Questions:**

Is there any paneltation to optimize/adjust the positions of four contral points in a curve?

As it will “scissored out” some part of a Gaussian point, will this affect the opacity of the whole point, and whether it will helps the point escape pruning.(pruning occurs when opacity < threshold)

**Limitations:**

The paper lacks of computational complexity of the proposed model. There is no explanation or discussion on the curves effect the densification procedure. Too many labels.

---

> ### Author Rebuttal · Authors · 2024-08-07
>
> >*Q1: Defines numerous labels. [...] It would be better to include a list of labels and their explanations. [...] label explanations under each equation.*
>
> **A1:** Thanks for your suggestion. Following it, (1) below, we formulate a list of labels (symbols) and their explanations. (2) Meanwhile, under each equation in our paper, we will fully explain all the labels used in that equation. For example, we will re-explain $\mu$, $r_1$, and $r_2$ under Eq. 4, and re-explain $P$ and $W$ under Eq. 5. We will also include a list of all labels and their explanations in the Appendix of our paper.
>
>
> |Label|Explanation|
> |---|---|
> |$\mu$|Center of the Gaussian|
> |$\Sigma$|Covariance matrix of the Gaussian|
> |$c_{SH}$|Spherical harmonic coefficients of the Gaussian|
> |$\alpha$|Opacity of the Gaussian|
> |$R$|Rotation matrix of the Gaussian|
> |$r_1$, $r_2$|First column and second column of the rotation matrix $R$|
> |$S$|Scale matrix of the Gaussian|
> |$\mu^{2D}$|Center of the projected Gaussian|
> |$\Sigma^{2D}$|Covariance matrix of the projected Gaussian|
> |$W$|Viewing transformation matrix|
> |$P$|Projective transformation matrix|
> |$J$|Jacobian of the affine approximation of the projective transformation|
> |$p$|Pixel|
> |$C(p)$|Color at pixel $p$|
> |$\omega_0$, $\omega_1$, $\omega_2$, $\omega_3$|Four control points of the cubic Bézier curve|
> |$B(t)$|Cubic Bézier curve function where $t$ is a random real number|
> |$B_{imp}(x, y)$|Implicit representation form of $B(t)$|
> |$M$|Hyperparameter representing the number of Bézier curve|
> |$c_{curve}$|Fifth attribute of the Gaussian|
> |$c_{curve}^{3D}$|$c_{curve}$ in 3D space coordinate system|
> |$c_{curve}^{2D}$|$c_{curve}$ in image plane coordinate system|
> |$g(p)$|Indicator function w.r.t. all the $M$ Bézier curves|
> |$g_{sc}(\omega_0, \omega_1, \omega_2, \omega_3; p)$|Single-curve indicator function|
> |$g_{sc}^0(p)$|Single-curve indicator function w.r.t. the first Bézier curve|
> |$L$|Loss function|
> |$\phi$|Desired value of $c_{curve}^{2D}[0, 0]$|
> |$S_{\phi}$|Set of all possible real number solutions for $\phi$|
> |$\widetilde{\phi}$|Solution in $S_{\phi}$ that is nearest to $c_{curve}^{2D}[0, 0]$|
> |$\widetilde{\phi_1}$|Solution in $S_{\phi}$ that is nearest to $c_{curve}^{2D}[0, 0]$ from its left side|
> |$\widetilde{\phi_2}$|Solution in $S_{\phi}$ that is nearest to $c_{curve}^{2D}[0, 0]$ from its right side|
> |$\epsilon$, $\epsilon_1$, $\epsilon_2$|Small numbers used to avoid the gradient exploding problem|
>
> >*Q2: Analysis of the computational complexity of the proposed method.*
>
> **A2:** Below, we show both the training time and the rendering time (inference time) of our proposed method, on the Tanks&Temples dataset on an RTX 3090 GPU.
>
> | Method | PSNR↑ | Training time | Rendering time (per image) |
> |---|---|---|---|
> | Mip-NeRF360 | 22.22 | 48 hours | 7.143s |
> | 2D Gaussian Splatting | 23.30 | 0.27 hour | 0.007s |
> | 3D Gaussian Splatting | 23.14 | 0.27 hour | 0.007s |
> | Ours (on 2D Gaussian Splatting) | 24.96 | 0.32 hour | 0.008s |
> | Ours (on 3D Gaussian Splatting) | 24.67 | 0.33 hour | 0.008s |
>
> As shown, though our framework achieves obviously better performance, it only brings a relatively small increase of training time. Besides, during inference, our framework can also achieve a competitive rendering time (speed) compared to the existing methods leveraging the conventional Gaussian Splatting technique and satisfy most real-time requirements. We will extend Tab. 11 in the paper for the table to also include the analysis of training time (besides the analysis of rendering time).
>
> >*Q3: How the curves affect the densification procedure. For example, will the new cloned/splitted points be assigned the same curve attribute?*
>
> **A3:** Yes. In the densification procedure of our framework, when a point (Gaussian) is cloned/splitted into two new points (Gaussians), similar to how other attributes such as the spherical harmonic coefficients and the opacity are assigned to the new points, we simply assign both the new points with the same curve attribute as the original point. We will discuss this in the paper.
>
> >*Q4: Is there any penalty to optimize/adjust the positions of four control points in a curve?*
>
> **A4:** Yes. In our framework for which we have maintained its differentiability, similar to the other attributes of the Gaussian, the positions of the four control points stored in the curve attribute are also learnable parameters. This means that, during training, based on the loss function as the penalty, the gradients would first backpropagate from the loss function to the positions of control points stored in the curve attributes. Leveraging such backpropagated gradients, the positions of control points would then be correspondingly optimized/adjusted to facilitate the accurate representation of the 3D scene.
>
> >*Q5: Will this affect the opacity of the whole point, and whether it will help the point escape pruning.*
>
> **A5:** (1) In Gaussian Splatting, the opacity of the whole point (Gaussian) that would be used during point (Gaussian) pruning is the opacity attribute $\alpha$. In our framework, to perform "scissoring out", we add a binary indicator function $g$ to the color blending function and this does not edit the opacity attribute $\alpha$ (as can be seen in Eq. 6 in the paper). (2) In other words, as long as "opacity $\alpha$ < threshold", the pruning process of a point (Gaussian) would always still occur in our framework, and our framework would not interfere this process. Thus, our "scissoring out" process would not help the point escape pruning.

---

### Author Rebuttal · Authors · 2024-08-07

We thank all reviewers for recognition of our contributions (Reviewer TofM: "an innovative framework", "a robust and sound pipeline"; Reviewer F99C: "a novel discontinuity-aware rendering pipeline", "addresses a very important problem", "the idea of using Bézier curves to define the shape is novel and sound"; Reviewer Kjj1: "significantly enhances the rendering results of scene boundaries", "innovatively, they propose using Bézier curves to draw continuous boundaries"; Reviewer ncnd: "a novel kernel function", "very reasonable and much needed by the community").

---

### Decision · Program_Chairs · 2024-09-25

**Decision:**

Accept (poster)

**Comment:**

This paper initially received mixed reviews. Reviewers raised concerns about the clarity of the manuscript and the role of the Bezier curve constraint on the quality of the results. The authors provided a rebuttal, and this paper was discussed at length by the reviewers and the AC. After the discussion, most reviewers agreed that the work is novel and that the additional results are sufficient to show the utility of the proposed Bezier curve constraint for improving reconstruction quality. The AC determined that the paper should be accepted. The authors should incorporate the results and clarifications from the rebuttal into the paper.